EMBO
Molecular Medicine

# Myopathy reversion in mice after restauration of mitochondrial complex I

Claudia V Pereira[1,†], Susana Peralta[1,†], Tania Arguello[1], Sandra R Bacman[1], Francisca Diaz[1] &
Carlos T Moraes[1,2,*] iD

## Abstract

**Myopathies are common manifestations of mitochondrial diseases. To investigate whether gene replacement can be used as an effective strategy to treat or cure mitochondrial myopathies, we have generated a complex I conditional knockout mouse model lacking NDUFS3 subunit in skeletal muscle. NDUFS3 protein levels were undetectable in muscle of 15-day-old smKO mice, and myopathy symptoms could be detected by 2 months of age, worsening over time. rAAV9-Ndufs3 delivered systemically into 15- to 18-day-old mice effectively restored NDUFS3 levels in skeletal muscle, precluding the development of the myopathy. To test the ability of rAAV9-mediated gene replacement to revert muscle function after disease onset, we also treated post-symptomatic, 2-month-old mice. The injected mice showed a remarkable improvement of the mitochondrial myopathy and biochemical parameters, which remained for the duration of the study. Our results showed that muscle pathology could be reversed after restoring complex I, which was absent for more than 2 months. These findings have far-reaching implications for the ability of muscle to tolerate a mitochondrial defect and for the treatment of mitochondrial myopathies.**

**Keywords** adeno-associated virus; complex I; gene therapy; mitochondrial myopathies; NDUFS3
**Subject Categories** Genetics, Gene Therapy & Genetic Disease; Musculoskeletal System

## Introduction

Mitochondrial diseases have a prevalence of approximately 1:4,300 in adults (Gorman *et al*, 2015) and about 1:11,000 in children younger than 6 years old (Darin *et al*, 2001). The heterogeneity of symptoms and the variable age of onset make these disorders challenging for therapy development. Therefore, despite the use of nutritional supplements, patients remain without effective treatments. Mitochondrial disorders can be caused by mutations either in mitochondrial DNA (mtDNA)- or in nuclear DNA (nDNA)-encoded genes. In contrast to the maternally inherited mtDNA, nDNA defects are transmitted in a Mendelian fashion and tend to be autosomal recessive (Zeviani & Di Donato, 2004).

Among many important cellular reactions, oxidative phosphorylation (OXPHOS) is a critical function of mitochondria. The OXPHOS system consists of four multi-subunit complexes (complexes I–IV) that constitute the electron transport chain (ETC) plus the FoF1-ATP synthase (complexes V) that uses the electrical and chemical gradients to drive the generation of ATP from ADP and inorganic phosphate (Pi) (Schon *et al*, 2012). The assembly of the OXPHOS system requires both mtDNA-encoded polypeptides and the synthesis and import of more than 100 nuclear-encoded proteins, synthesized on cytosolic ribosomes (Wiedemann & Pfanner, 2017).

NADH-ubiquinone oxidoreductase (complex I, CI) deficiency is amongst the most frequent defects of the OXPHOS system and has been associated with a wide variety of clinical symptoms. Mutations have been found in either mitochondrial or nuclear DNA-encoded subunits, which resulted in impaired catalytic efficiency or inability to assemble the holoenzyme complex (Antonicka *et al*, 2003). However, mutations in nuclear genes coding for CI subunits are the most common cause of isolated CI enzyme deficiencies (Distelmaier *et al*, 2009; Rodenburg, 2016). Due to their high energy requirements, organs such as the skeletal and cardiac muscles, and the brain are often affected. Hypotonia (weak muscle tone) and ataxia (problems with equilibrium and movement coordination) are commonly observed in these patients (Arii & Tanabe, 2000; Antonicka *et al*, 2003; Finsterer, 2008; Distelmaier *et al*, 2009). In addition, some individuals develop eye movement disorders or optic nerve degeneration. Most affected children with CI deficiency present Leigh-like syndrome (LS), which is a progressive disease associated with psychomotor retardation, brainstem dysfunction, seizures, failure to thrive, muscular hypotonia, dystonia, abnormal eye movements, and lactic acidosis (Rahman *et al*, 1996; Finsterer & Zarrouk-Mahjoub, 2017). In addition, CI defects may also cause isolated hypertrophic cardiomyopathy (HCM) (Loeffen *et al*, 2001;

---

1 Department of Neurology, University of Miami Miller School of Medicine, Miami, FL, USA
2 Department of Cell Biology, University of Miami Miller School of Medicine, Miami, FL, USA
*Corresponding author. Tel: +1 305 243 5858; E-mail: cmoraes@med.miami.edu
†These authors contributed equally to this work

Benit *et al*, 2003; Fassone *et al*, 2011) or Leber's hereditary optic neuropathy (LHON; Kirches, 2011).

The L-shaped CI is by far the largest of the ETC complexes with ~ 980 kDa, encompassing 45 subunits in mammals (Mimaki *et al*, 2012). The core bioenergetic functions of CI are executed by 14 central subunits that are conserved from bacteria to humans. Seven of the hydrophobic subunits are encoded by the mtDNA (ND1, ND2, ND3, ND4, ND4L, ND5, and ND6) and belong to the proton translocase P module. The other seven subunits are more hydrophilic and encoded by the nDNA, and these subunits are in the N and Q modules (Loeffen *et al*, 2000; Vinothkumar *et al*, 2014). In addition, there are 31 nDNA-encoded subunits referred to as "supernumerary" or "accessory" and their actual function is still unknown (Mimaki *et al*, 2012; Wirth *et al*, 2016; Zhu *et al*, 2016). The CI-N module, responsible for the oxidation of NADH, includes at a minimum the NDUFV1, NDUFV2, and NDUFS1 subunits. And the Q module, responsible for the electron transfer to ubiquinone, includes at a minimum of the NDUFS2, NDUFS3, NDUFS7, and NDUFS8 subunits (Mimaki *et al*, 2012; Wirth *et al*, 2016). Furthermore, it was described that the 39 kDa subunit (NDUFA9) is attached to the 20 kDa subunit NDUFS7 (PSST) and to the 30 kDa subunit (NDUFS3) in the hydrophilic arm (Zhu *et al*, 2016). To date, some mouse models for isolated CI deficiency in mitochondrial CI subunits have been reported, including NADH-ubiquinone oxidoreductase subunit S4 (*Ndufs4*) (Kruse *et al*, 2008; Ingraham *et al*, 2009; Quintana *et al*, 2010, 2012; Chen *et al*, 2017), S6 (*Ndufs6*) (Ke *et al*, 2012), and NADH-ubiquinone oxidoreductase 1 alpha subcomplex subunit 5 (*Ndufa5*) (Peralta *et al*, 2014), all of which are "supernumerary" subunits. In addition, a mutant mtDNA mouse model for *MT-ND6* CI subunit for Leber's hereditary optic neuropathy (LHON) was also created and explored (Lin *et al*, 2012).

The NADH-ubiquinone oxidoreductase core subunit S3 (NDUFS3) belongs to the "minimal assembly" required for enzyme formation and activity (Vogel *et al*, 2007). Compound heterozygosity for *Ndufs3* mutations (T145I and R199W) has been reported in one patient, and it was associated with late-onset LS, optic atrophy, and CI deficiency (Benit *et al*, 2004). A different compound heterozygosity (R140W and R199W) has been associated with early-onset Leigh syndrome and severe reduction in CI levels (Lou *et al*, 2018a,b). In addition, another patient with development delay, encephalopathy, myopathy, and lactic acidosis with homozygosity for the R199W mutation has been reported (Haack *et al*, 2012).

In the present study, we have generated a novel CI-conditional knockout (smKO) mouse model by deleting the *Ndufs3* gene specifically in the skeletal muscle. We show that our model can mimic the myopathy phenotype observed in patients with mitochondrial disease. After thoroughly characterizing the *Ndufs3*-smKO mice, we treated two different age groups with recombinant adeno-associated virus subtype 9 (rAAV9) expressing the mouse *Ndufs3*. The first group of animals received a single retro-orbital rAAV9-*Ndufs3* injection at post-natal days 15–18 and the second group, at 2 months of age. Our results showed an apparent complete recovery of muscle function and biochemical features in both groups of pre- and post-symptomatic mice. Importantly, this study implies that a wide temporal therapeutic window for gene therapy is possible for mitochondrial myopathies.

# Results

## Creation and characterization of a skeletal muscle-specific Ndufs3 smKO mice

To knock out *Ndufs3* in skeletal muscle, mice homozygous for a floxed *Ndufs3* (*Ndufs3*$^{f/f}$, Appendix Fig S1A and B) were bred with transgenic mice expressing Cre recombinase under the Myosin light chain 1 promoter (*Myl1* or *Mlc1f*; Bothe *et al*, 2000). The *Mlc1f* gene is expressed strongly during muscle development (Lyons *et al*, 1990) and is selective to differentiated skeletal muscle, giving rise to a *Ndufs3* smKO mice in the skeletal muscle (*Ndufs3*$^{f/f}$*Mlc1f*-Cre$^{+/-}$). *Ndufs3*$^{f/f}$*Mlc1f*-Cre$^{-/-}$ or *Ndufs3*$^{f/W}$*Mlc1f*-Cre$^{-/-}$ were used as wild-type (WT$^{flx}$) control animals (Appendix Fig S1B). The mice were genotyped as illustrated in Appendix Fig S1C. The absence of NDUFS3 protein was confirmed by Western blot analysis in homogenates of muscle quadriceps from 15 days and 1-month-old animals (Fig 1A). *Ndufs3* smKO mice were born at Mendelian ratios and showed similar body weight when compared to WT$^{flx}$ at early ages (Fig 1B). At the age of 3 months, smKO males showed a significant decrease in body weight when compared to wild-type mice, whereas females showed a significant decrease starting at 5 months of age (Fig 1B). *Ndufs3* smKO mice lost weight progressively (Fig 1C) and died prematurely (Fig 1D). We found that 50% of *Ndufs3* smKO mice (males or females) died by 8 months of age (Fig 1D). No significant gender differences were observed in the lifespan of smKO mice (Fig 1D).

To characterize the motor phenotype of these mice, we used several behavioral tests in groups of male mice. *Ndufs3* smKO mice showed decreased scores in the ambulatory activity cage test starting at 1 month of age (Fig 1E). At the age of 2 months, *Ndufs3* smKO mice moved significantly less than wild-type mice throughout the night (Fig 1E). The hypoactivity manifested by *Ndufs3* smKO mice continued to worsen with time. Treadmill (Fig 1F) and rotarod tests (Fig 1G) were used to analyze motor skills and coordination. The latency to fall in the treadmill was severely decreased in the *Ndufs3* smKO mice from 2 months onward, demonstrating exercise intolerance (Fig 1F). In agreement, motor activity measured in the rotarod was also decreased at 2 months of age and worsened with time (Fig 1G). Furthermore, we evaluated motor activity of the *Ndufs3* smKO mice in an open field test and found reduced stereotypical time at 2 months of age (Appendix Fig S2B). These results indicate that the lack of NDUFS3 in skeletal muscle caused a severe deterioration of the locomotor activity with a clear onset between 1 and 2 months, progressing to a premature death.

## Lack of NDUFS3 induces muscle degeneration accompanied by increased mitochondrial proliferation and serum lactic acidosis

Muscles of *Ndufs3* smKO mice at 1 month or more were darker and had an altered consistency compared with the muscles from wild-type animals (Fig 2A and Appendix Fig S2C). Absolute muscle weight was decreased at 6 months of age in the *Ndufs3* smKO mice, and the differences were exacerbated at 8 months (Fig 2A and Appendix Fig S2D). The severe muscle loss also had an impact on the overall weight of the animals, as these differences were not observed when normalized to total body weight (Appendix Fig S2E). Differences in weight and color were not significant in the

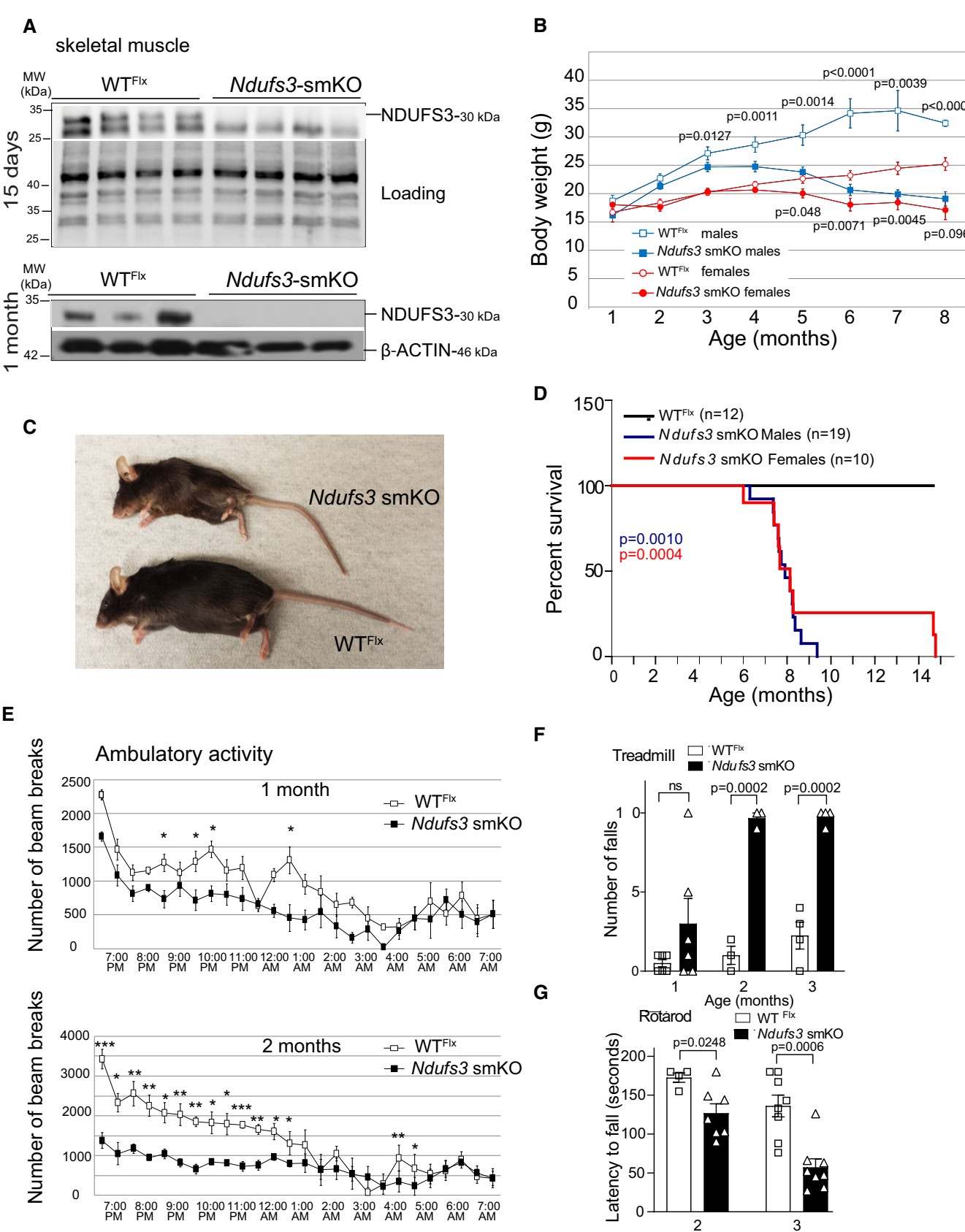

Figure 1.

Figure 1. Creation and characterization of skeletal muscle-specific *Ndufs3* KO (smKO) mice.

A   Western blot analysis of NDUFS3 protein levels in homogenates from quadriceps muscle of 15-day-old (upper panel) and 1-month-old animals (lower panel).
B   Body weight comparison over time of *Ndufs3* smKO male mice (filled blue squares; $n = 14$) and age-matched wild-type (WT$^{Flx}$; $n = 15$) males (empty blue squares). *Ndufs3* smKO females (filled red circles) showed similar results ($n = 10$ for smKO and $n = 12$ for WT$^{Flx}$ group). Data represents means ± SEM. *P* values were calculated by Student's *t*-test.
C   Representative image of a smKO male *Ndfu3s3*-smKO mouse and a WT$^{Flx}$ littermate of 6 months old showing decreased body weight.
D   Survival curve of *Ndufs3* smKO female mice (red line) and males (blue line). *P* values were calculated using Log-rank (Mantel-Cox) test; $P = 0.0010$ for males ($n = 19$), and $P = 0.0004$ for females ($n = 10$).
E   Nocturnal ambulatory activity of *Ndufs3 smKO* and age-matched wild-type littermates from 1- and 2-month-old male mice. Data represents means ± SEM. *P* values were calculated by Student's *t*-test. *$P < 0.05$, **$P < 0.01$, and ***$P < 0.0001$.
F   The number of falls in the treadmill of WT$^{Flx}$ and *Ndufs3* smKO male mice, starting at 2 months ($n = 8$). Bars represent means ± standard error (SEM). *P* values were calculated by Student's *t*-test to determine the level of statistical difference. n.s.—not significant.
G   Rotarod performed by *Ndufs3* smKO and wild-type male mice ($n = 8$). Bars represent means ± standard error (SEM). *P* values were calculated by Student's *t*-test to determine the level of statistical difference.

slow twitch soleus as *Mlc1f* gene is expressed mostly in fast twitch-rich skeletal muscles (Lyons *et al*, 1990; Appendix Fig S2C).

The muscle fibers from *Ndufs3* smKO were smaller compared to wild types with increased number of fibers presenting central nuclei, at 8 months (Fig 2B and C). In addition, we observed increased cytochrome *c* oxidase (COX) activity by histochemical staining in fresh-frozen quadriceps sections from *Ndufs3* smKO (Fig 2B). Likewise, succinate dehydrogenase (SDH) activity staining was also increased (Fig 2B). Electron microscopy analyses in quadriceps (Appendix Fig S3A) showed abnormal mitochondria in the smKO mice muscles. In the *Ndufs3* smKO, mitochondria appeared clustered, with condensed cristae compared to wild-type samples (Appendix Fig S3A). Moreover, we observed lipid inclusions in the *Ndufs3* smKO quadriceps (Appendix Fig S3B), but not in controls. Taken together, our results show that the ablation of *Ndufs3* in muscle induced a mitochondrial myopathy characterized by muscle loss and abnormal mitochondrial structure.

Furthermore, serum lactate levels from *Ndufs3* smKO mice at 3, 6, and 8 months were dramatically increased, demonstrating that *Ndufs3* smKO mice suffered from severe lactic acidosis (Fig 2D), a common feature in patients with mitochondrial myopathies (Rowland *et al*, 1991).

**Ndufs3 smKO mice had decreased CI expression and activity**

NDUFS3 forms a sub-assembly complex together with NDUFS2, NDUFS7, and NDUFS8 during the early stages of complex I assembly (Mimaki *et al*, 2012; Guerrero-Castillo *et al*, 2017); therefore, we expected that loss of NDUFS3 would impair CI assembly. Indeed, we found a marked decrease in CI in-gel activity detected by BN-PAGE in quadriceps homogenates from *Ndufs3* smKO of 2 and 4 months of age (Fig 2E). In agreement with the histochemical findings, the absence of CI in skeletal muscle was accompanied by an increase in complex IV activity, clearly visible at 4 months (Fig 2E).

PGC-1α, a co-transcriptional activator regulating mitochondrial biogenesis, was increased in the quadriceps of 2- and 4-month-old smKO mice (Fig 2F–H). The amount of the mitochondrial complex IV subunit, COX1 in smKO was three times higher than in wild types at the age of 2 months, and 10 times higher at 4 months (Fig 2F–H). In fact, increased COX1 expression in the smKO samples was already observed at 1 month (Appendix Fig S2F). VDAC-1 levels were similar at 2 months but doubled at 4 months in smKO mice (Fig 2F–H). The amount of the complex II subunit (SDHA) was also already significantly higher at 2 months in the smKO group (Fig 2F–H).

Altogether, these results showed that the absence of CI subunit NDUFS3 induced a robust mitochondrial proliferation, likely as a compensatory mechanism.

**rAAV9 retro-orbital injections prevented the myopathy development in young Ndufs3 smKO mice**

Recombinant adeno-associated virus serotype 9 (rAAV9) containing the wild-type mouse *Ndufs3* cDNA or eGFP was systemically delivered to 15- to 18-day-old smKO mice via retro-orbital injections, as described (Yardeni *et al*, 2011; Long *et al*, 2016). Gene expression was driven by the cytomegalovirus (CMV) promoter (Fig 3A). Our study consisted of a single injection in two different groups of smKO mice. The smKO control group (KO-GFP) was administered rAAV-eGFP, whereas the other group was administered rAAV9-*Ndufs3* (KO-NDUFS3) viral genomes. The study protocol is summarized in Fig 3B. The KO-GFP mice were visibly smaller and weaker than KO-NDUFS3 mice at 7 months (Fig 3C). Our results showed that the body weight of rAAV9-*Ndufs3*-injected smKO mice was similar to WT$^{Flx}$ animals at all ages studied, even 5 months after the injection (Fig 3D and E). However, the body weight of the smKO injected with rAAV-*eGFP* was lower than WT$^{Flx}$ mice, starting at 4–5 months (Fig 3D and E). The treadmill results showed a completely normal phenotype from 1.5 months after treatment and onward, for both males and females (Fig 3F). The rAAV9-*eGFP*-injected mice showed signs of exercise intolerance at 2 months, and their phenotype progressively deteriorated over time. The four-limb hang test showed a significant improvement of the KO-NDUFS3 mice 2.5 months after injections, whereas the KO-GFP mice showed a progressive muscle weakness at 2 months (Fig 3G). In addition, the muscle weight of KO-NDUFS3 mice was similar to WT$^{Flx}$, in contrast to KO-GFP mice, which showed significant atrophy, at 6 months (Fig 3H). Analysis of the behavioral data showed no significant differences between genders (Appendix Fig S4A and B); thus, the data were combined.

We have allowed some mice to age and observed that rAAV9-*Ndufs3* injections markedly extended survival (Fig 3I). Two of these mice were sacrificed at 15 months for protein expression analyses by Western blot, with no overt signs of myopathy (Appendix Fig S5E and F). The cause of death of one of the injected mice is unknown but its weight was normal (24 g, similar to WT$^{Flx}$) before its death at 8 months. The other two KO-NDUFS3 mice also died for apparently unrelated reasons (no evidence of myopathy) around 6 months. On the other hand, all KO-GFP mice died before reaching

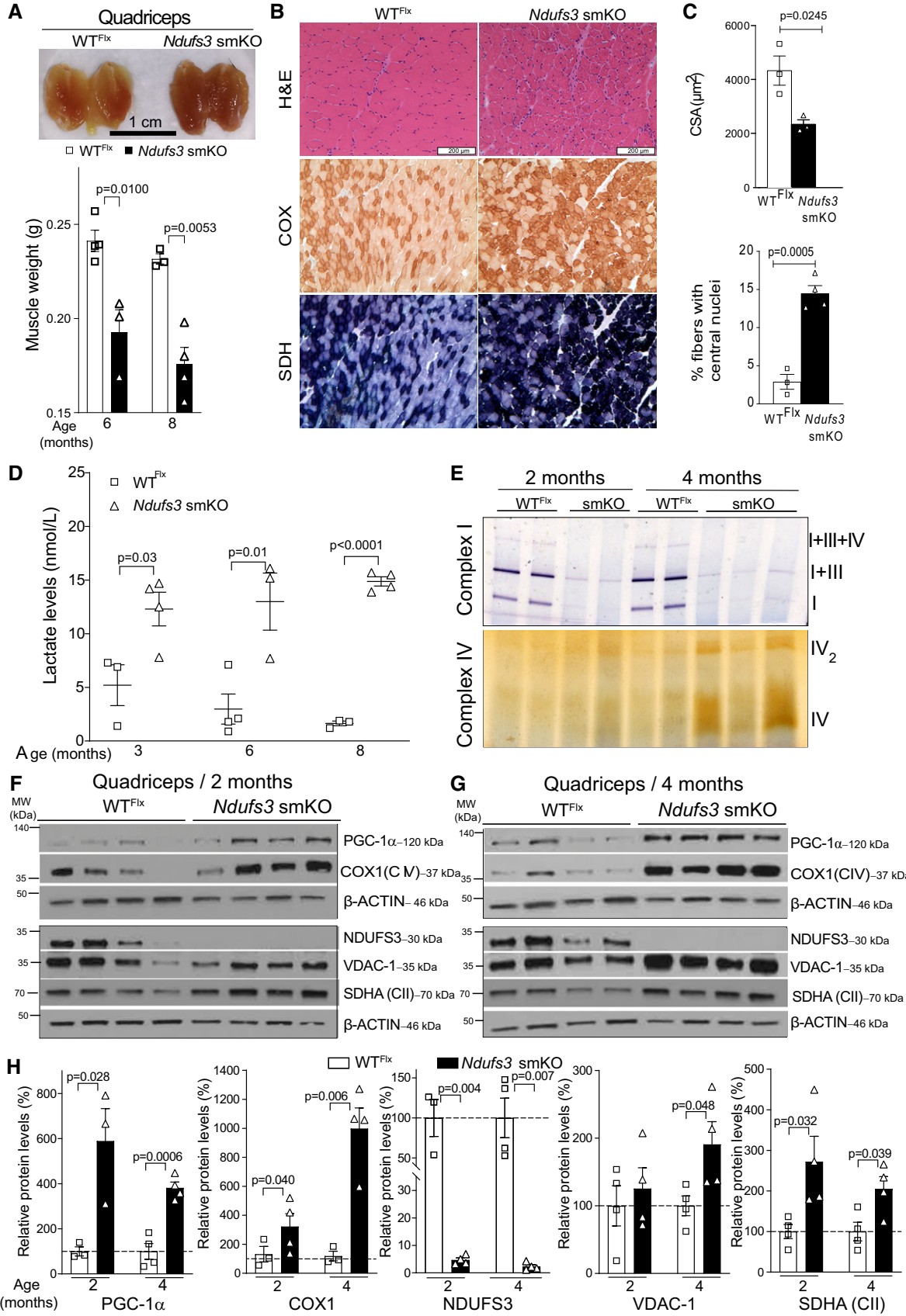

**Figure 2.**

**Figure 2.  Lack of NDUFS3 in skeletal muscle is associated with a mitochondrial myopathy.**

A   Representative image of quadriceps from WT$^{Flx}$ and *Ndufs3* smKO animals of 8 months of age. Muscle weight from *Ndufs3* smKO males (black bar) and wild-type mice (white bar) from 6 and 8 months of age (n = 5 each group). Bars represent means ± standard error (SEM). P values were determined by Student's *t*-test.

B   Representative H&E, COX, and SDH staining of quadriceps from WT$^{Flx}$ and *Ndufs3-Mlc1f* smKO at 8 months old. Scale bar, 200 μm.

C   Cross-sectional area (CSA) of quadriceps muscle fibers. A minimum of 100 fibers/sample were analyzed (n = 3/genotype). The percentage of central nuclei in skeletal muscle fibers of quadriceps muscle of WT$^{Flx}$ and *Ndufs3* smKO at 8 months (n = 3 mice for WT$^{Flx}$ and n = 4 mice for *Ndufs3* smKO). Bars represent means ± SEM. P values were calculated by Student's *t*-test.

D   Serum lactate levels of WT$^{Flx}$ and *Ndufs3*-smKO animals at different ages. *Ndufs3* smKO male mice (filled black squares) showed increased levels compared to matched WT$^{Flx}$ males (empty black squares). Error bars represent SEM. P values were determined by Student's *t*-test.

E   BN-PAGE in-gel activities of complex I and IV measured in homogenates from quadriceps of wild-type and *Ndufs3-Mlc1f* smKO animals at 2 and 4 months.

F, G   Western blots of muscle homogenates (quadriceps) of WT$^{Flx}$ and *Ndufs3-Mlc1f* smKO animals at different ages (2 and 4 months, respectively) using antibodies against PGC-1α, COX1 (complex IV subunit), NDUFS3, VDAC1, SDHA (complex II subunit), and β-actin.

H   Quantification of the Western blots in panels (F and G). Bars represent means ± SEM of n = 4 for each group. P values were determined by Student's *t*-test.

8 months of age (Fig 3I), emaciated and weak (depicted in Fig 3C), consistent with the previous observations.

### The smKO rAAV-Ndufs3-injected mice showed restored CI, mitochondrial markers, and serum lactate levels

Protein levels in muscle homogenates (quadriceps) of all mouse groups were evaluated by Western blot at 6 months (Fig 4A and B); a time-point selected due to the smKO short lifespan. NDUFS3 (30 kDa) protein levels were restored in both quadriceps and gastrocnemius (Fig 4A and B, Appendix Fig S5A and B). SDHA (complex II, nDNA-encoded subunit) protein expression was increased in the smKO-GFP mice, whereas in smKO-NDUFS3 mice, it was similar to WT$^{Flx}$ levels (Fig 4A and B). Comparable results were found for COX1 (complex IV, mtDNA-encoded subunit) (Fig 4A and B). As expected, GFP expression in quadriceps of KO-GFP mice was confirmed 6 months after injection (Fig 4A). The NDUFS3 expression and GFP expression were also analyzed in other tissue homogenates (Appendix Fig S5A and B). GFP expression was found in other tissues besides skeletal muscle, such as heart, liver, and, in some mice, brain (Appendix Fig S5A and B).

The smKO mice injected with GFP showed a marked decrease of CI and supercomplexes assembly, demonstrated by blue native Western blots (Fig 4C and D). We observed a remarkable recovery of CI levels in BN-PAGE experiments after rAAV9-*Ndufs3* injections in both females (Fig 4D) and males (Appendix Fig S4C). Complexes IV and II were also analyzed and once again showed similar to WT$^{Flx}$ expression levels in the KO-NDUFS3 samples, whereas KO-GFP showed increased levels of these complexes under native-gel conditions (Fig 4C and D). CI in-gel activity of KO-NDUFS3 homogenates also showed complete normalization of the smKO biochemical phenotype (Fig 4E and F). In-gel CIV activity in KO-NDUFS3 female quadriceps was similar to the ones observed in WT$^{Flx}$ mice (Fig 4E and F). The KO-GFP mice showed increased CIV in-gel activity (Fig 4E and F). Similar results were obtained with males (Fig 4E and Appendix Fig S4D).

Histological analysis showed numbers of central nuclei in skeletal muscle fibers of KO-NDUFS3 similar to WT$^{Flx}$ mice, evaluated by H&E staining at 6 months (Appendix Fig S4E and F). Furthermore, we observed increased COX/SDH tissue activity staining in KO-GFP mice (Appendix Fig S4C), whereas the KO-NDUFS3 mice showed normal (similar to WT$^{Flx}$) histochemical activities (Appendix Fig S4C). mtDNA levels in the KO-GFP samples were increased but restored to WT$^{Flx}$ levels in the smKO-NDUFS3 (Fig 4G). The

consistent empirical observation of the dark-red colored muscle in the smKO mice is related to the increased mitochondrial levels present in this mouse model (inset in Fig 4G). The KO-NDUFS3 quadriceps exhibited a normal appearance ~ 6 months after rAAV9-*Ndufs3* injection (Fig 4G). These results, together with the Western blot data, indicate mitochondrial proliferation with increasing levels of OXPHOS proteins, a likely compensatory mechanism in the absence of CI. In addition, CI (complex I)/CS (citrate synthase) spectrophotometric activity assays were performed in quadriceps homogenates of the three groups of animals. KO-GFP mice samples showed a 50% decrease of CI/CS activity, whereas KO-NDUFS3-injected mice showed a significant improvement of CI/CS activity (Fig 4H). High levels of lactate were present in the serum of 6-month-old KO-GFP mice but not in the KO-NDUFS3 mice (Fig 4I).

We have also determined the viral genomes (vg) number in DNA samples from quadriceps, gastrocnemius, heart, liver, and brain of KO-NDUFS3 mice and non-injected animals. The results were identical with two independent TaqMan probes, showing expression in all tissues analyzed. Brain showed the lowest and liver the highest expression. Skeletal muscle and heart also showed robust expression (Appendix Fig S5C and D).

To further explore the extent of rAAV9-*Ndufs3* expression in skeletal muscle of the KO-NDUFS3 animals that were allowed to age to 15 months, Western blot analysis was performed (Appendix Fig S5E and F). Interestingly, all muscles analyzed still expressed NDUFS3 (Appendix Fig S5E and F). The KO-GFP animals were 6 months old when the samples were collected, as none of the rAAV9-e*GFP*-injected mice survived longer than 8 months, and most were dead by 6–7 months. These results showed that rAAV9-*Ndufs3* administration was able to prevent the onset of the myopathy and completely normalize the biochemical phenotype associated with NDUFS3 absence in muscles of the smKO mice.

### rAAV9 gene replacement therapy reverted the myopathy phenotype of Ndufs3 smKO symptomatic mice

To test whether our gene therapy approach could revert the myopathy phenotype after disease onset, retro-orbital injections were performed in symptomatic smKO mice (2-month-old) with either rAAV9-*Ndufs3* or rAAV9-*eGFP*. A similar experimental strategy was applied for this group of older mice, as illustrated in Fig 5A. The smKO mice were tested before the injections, and all showed poor performance in the treadmill and in the hang test (Fig 5B and C). KO-NDUFS3-injected mice showed a significant recovery in

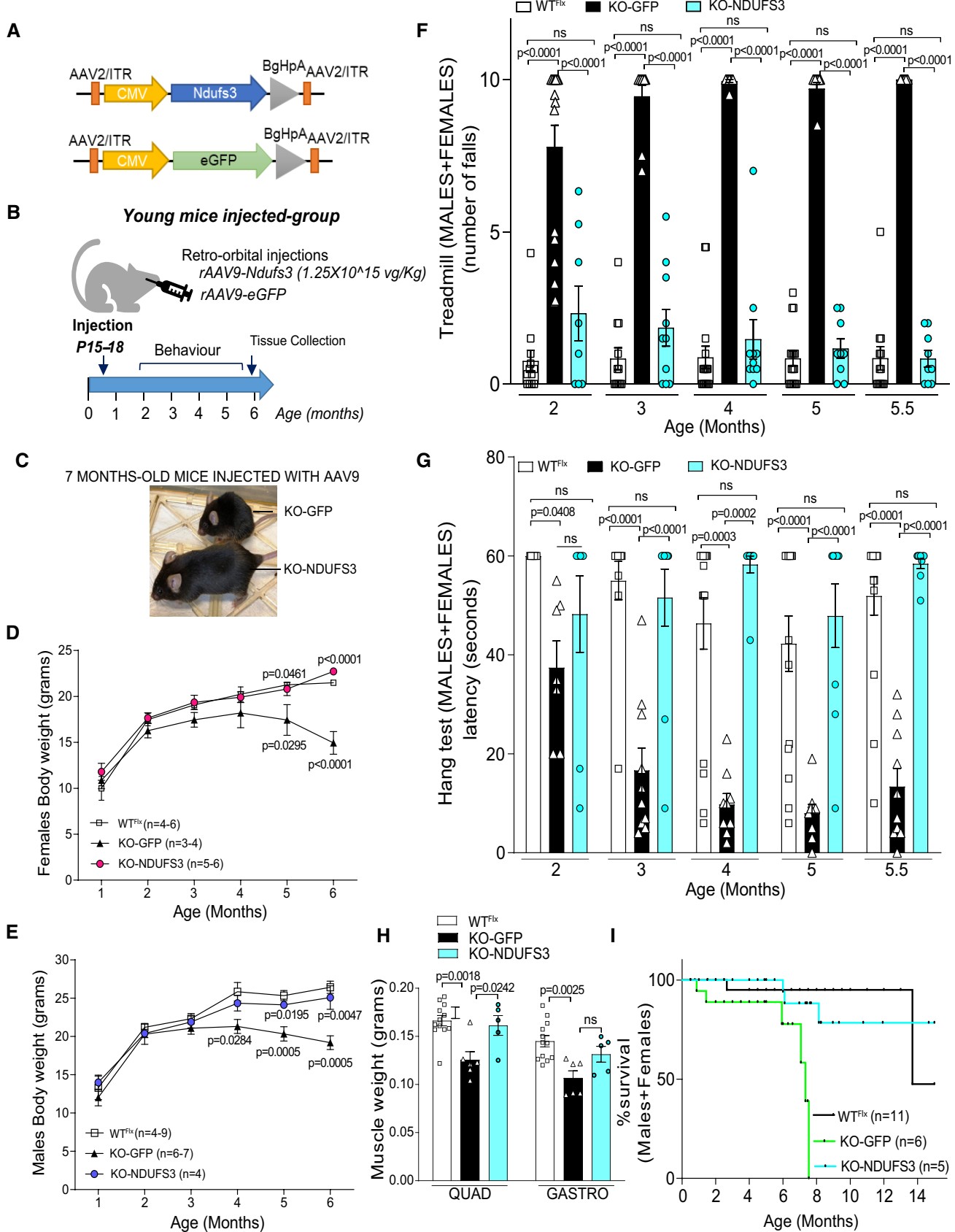

**Figure 3.**

**Figure 3.  Replacement of NDUFS3 in 15- to 18-day-old smKO mice prevented the development of myopathy.**

A   Representation of the vectors used in this study. WT^Flx *Ndufs3* cDNA was expressed from rAAV9-*Ndufs3*. An rAAV9-*eGFP*-expressing viral preparation was used as a control.
B   Experimental design used for rAAV9 administration and analyses.
C   The KO-GFP mouse was visibly smaller and weaker than a KO-NDUFS3 mouse, at 7 months.
D   Females body weight of mice injected with rAAV9-*Ndufs3* (heretofore referred to as KO-NDUFS3) were similar to WT^Flx mice body weight and significantly higher than the rAAV9-*eGFP*-injected mice (KO-GFP) starting at 5 months. Statistical analysis was performed by one-way ANOVA followed by Tukey post-test. Error bars represent ± SEM.
E   Males body weight of KO-NDUFS3 mice were similar to WT^Flx mice and significantly different from KO-GFP from 4 months onward. Statistical analysis was performed by one-way ANOVA followed by a Tukey post-test. Error bars represent ± SEM.
F   Treadmill performance was significantly improved in the KO-NDUFS3 mice when compared to the KO-GFP, from 2 months onward. Statistical analysis was performed by one-way ANOVA followed by Bonferroni post-test. Error bars represent ± SEM. WT^Flx *n* = 12–16, KO-GFP *n* = 9–18, KO-NDUFS3 *n* = 9–10.
G   Four-limb strength was evaluated by the grid hang test. KO-NDUFS3 mice performed significantly better starting at 3 months, when compared to the KO-GFP mice. Statistical analysis was performed by one-way ANOVA followed by Bonferroni post-test. Error bars represent ± SEM. WT^Flx *n* = 7–16, KO-GFP *n* = 7–10, KO-NDUFS3 *n* = 8–10.
H   Quadriceps and gastrocnemius wet weight of 6-month-old mice was similar between WT^Flx and KO-NDUFS3 mice. Quadriceps weight was significantly increased in KO-NDUFS3 mice when compared to KO-GFP injected. Statistical analysis was performed by one-way ANOVA followed by Bonferroni post-test. ns—not significant. Error bars represent ± SEM. WT^Flx *n* = 12, KO-GFP *n* = 6, KO-NDUFS3 *n* = 5.
I   Survival curve of *Ndufs3* smKO mice injected with rAVV9-*Ndufs3* (blue line) and with rAVV9-eGFP (green line). WT^Flx mice are depicted in black. rAAV9-*Ndufs3* injections extended the survival of the smKO mice. *P* values were calculated using Log-rank (Mantel-Cox) test; *P* = 0.0016 (KO-NDUFS3 vs. KO-GFP); *P* = 0.0014 (KO-GFP vs. WT^Flx); *n*—indicates the number of animals used per group.

treadmill performance 3 months after the injections (Fig 5B). Accordingly, KO-NDUFS3 muscle function measured with the four-limb hang test was also improved, 3.5 months after the injection (Fig 5C). Muscle weight was increased in KO-NDUFS3 when compared to KO-GFP mice, but only significantly different in gastrocnemius (Fig 5D). Furthermore, immunohistochemical analysis of NDUFB8 (as marker for CI) expression levels and of COX1 were consistent with a recovery in the KO-NDUFS3 mice (Fig 5E). The KO-GFP did not express NDUFB8 and appeared to have increased COX1 levels, whereas the KO-NDUFS3 mice had recovered CI and similar to WT^Flx mice COX1 expression (Fig 5E). Furthermore, the COX/SDH staining in the KO-NDUFS3 mice quadriceps was comparable to the one in WT^Flx (Fig 5F). We note that very high titers of rAAV9-*Ndufs3* were required to obtain muscle expression in adult mice after systemic injections. Multiple attempts to use lower titers in adult mice (up to $10^{14}$ vg/kg) resulted in no detectable expression of the transgene.

**Complex I assembly, mtDNA, and serum lactate levels were restored after rAAV9-Ndufs3 treatment of post-symptomatic mice**

KO-NDUFS3 mice injected at 2 months showed a rescue of NDUFS3 expression, accompanied by a recovery of the assembly into supercomplexes, by 6 months (Fig 6A–C, respectively). PGC-1α expression, which is increased in the KO-GFP mice, was normalized in the KO-NDUFS3. COX1 (CIV), Core 2 (CIII), and SDHA (CII) expression levels in KO-NDUFS3 mice were vastly reduced when compared to the KO-GFP mice (Fig 6A and B). NDUFS3 expression was evaluated in gastrocnemius and other tissues such as the heart, liver, and brain (Appendix Fig S6A and B). The vg copies of AAV9-NDUFS3 were also evaluated, as previously described. All tissues analyzed showed the presence of viral genome copies with both probes used (Appendix Fig S6C and D). NDUFS3 expression and GFP expression were further observed in the diaphragm, TA and arm muscles of KO-NDUS3 and KO-GFP mice, respectively (Appendix Fig S6E).

Moreover, recovery of CI levels and assembly into supercomplexes was detected in BN-PAGE experiments, after rAAV9-*Ndufs3*

injections in KO-NDUFS3 quadriceps (Fig 6C and D). Complexes IV and II levels from KO-NDUFS3 samples were also analyzed and were similar to WT^Flx, whereas KO-GFP showed increased levels of these complexes. CI in-gel activity showed a clear rescue of activity in KO-NDUFS3 mice (Fig 6E and F). CIV in-gel activity was decreased in KO-NDUFS3 quadriceps when compared to KO-GFP (Fig 6E and F). Spectrophotometric CI/CS activity of smKO injected with rAAV9-*Ndufs3* was not different from WT^Flx mice (Fig 6G). Total mtDNA levels in KO-NDUS3 homogenates were also similar to WT^Flx (Fig 6H). Finally, the serum lactate levels were significantly decreased in the KO-NDUFS3 serum samples when compared to the KO-GFP levels, 4 months after injections (Fig 6I).

## Discussion

The conditional absence of NDUFS3 in the skeletal muscle of our CI-deficient mouse model resulted in the development of a progressive myopathy that reduced lifespan to approximately 6–8 months. The cause of death was not determined, but mice were profoundly emaciated and unable to move normally around the cage. The progression of the disease phenotype was accompanied by a remarkable reduction of CI levels. Similarly, a patient carrying *Ndufs3* mutations (T145I and R199W) also showed a decrease in CI activity in muscle (Benit *et al*, 2004; Haack *et al*, 2012). Noteworthy, we have observed an increase in the levels of several mitochondrial markers, including: SDHA, VDAC, COX1, and mtDNA. This apparent compensatory mechanism could be explained by the observed increase in PGC-1α protein levels, a master regulator of mitochondrial biogenesis (Finck & Kelly, 2006). In accordance, mitochondrial proliferation is observed in both mitochondrial diseases and aging (Wallace *et al*, 1995). In other animal models, skeletal muscle mitochondria have been shown to proliferate in response to increased oxidative demand associated with the overexpression of human lipoprotein lipase (LPL) (Levak-Frank *et al*, 1995). Likewise, the Ant1 (adenine nucleotide translocator 1)-deficient mouse presented significant skeletal muscle mitochondrial proliferation (Graham *et al*, 1997). Other skeletal muscle KO models, such as the AIF-deficient mice (muscle-

                                      

specific apoptosis-inducing factor KO, MAIFKO) developed muscle atrophy and reduced CI expression, elevated serum lactate levels but did not show any increase in the activity of CII or CIV (Joza *et al*,

2005). Furthermore, the COX10 smKO mouse model showed mild mitochondrial proliferation, and only in older smKO animals (Diaz *et al*, 2005). This may indicate that the observed robust

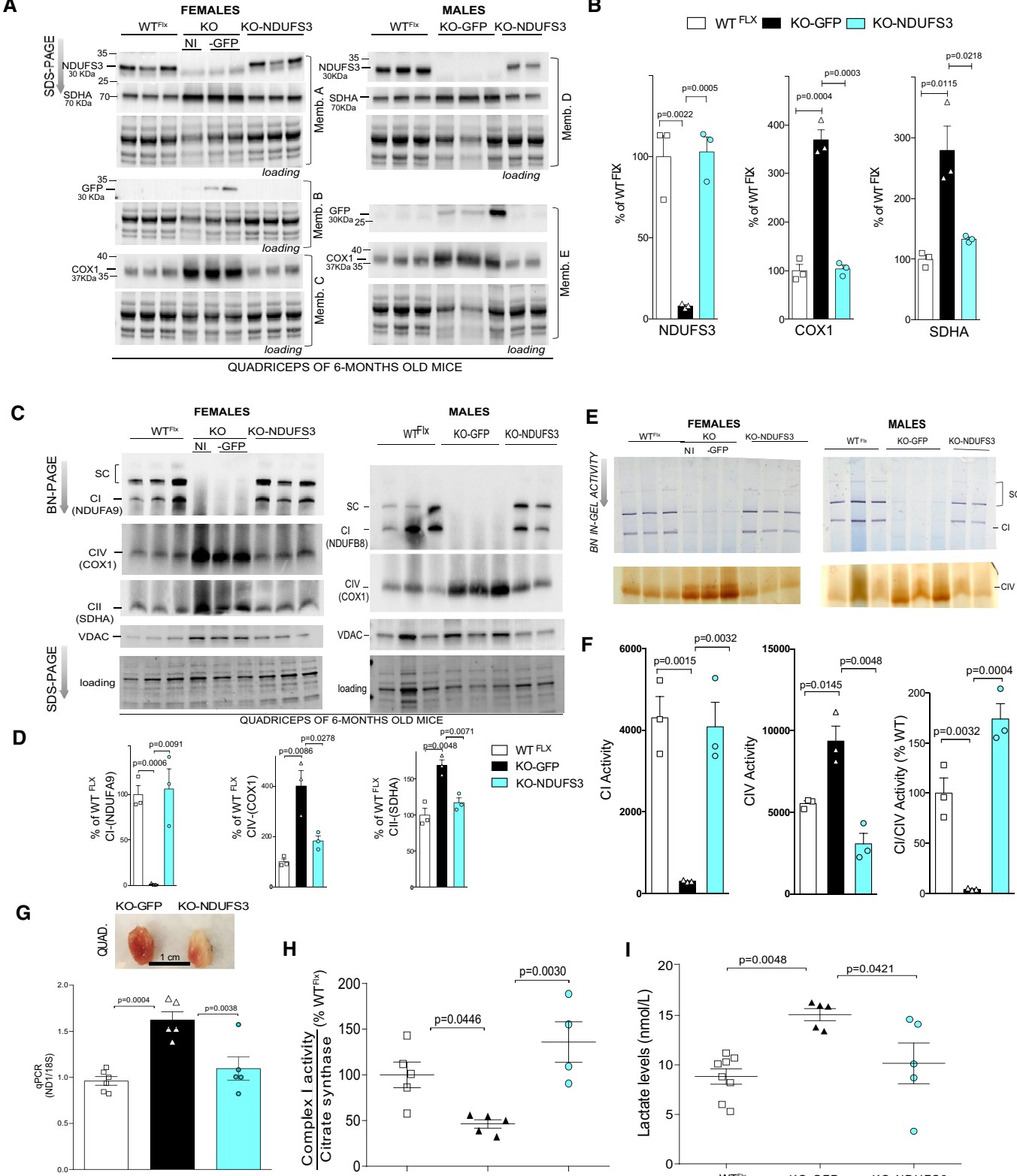

**Figure 4.**

**Figure 4. NDUFS3 replacement (at P15–18) normalized the levels of lactate and mitochondrial markers.**

A   Western blots of 6-month-old quadriceps of WT^Flx-, KO-, and KO-NDUFS3-injected mice at 15–18 days old. GFP expression was observed only in the KO-GFP mice. Protein loading was obtained with stain-free gel technology.

B   Quantification of NDUFS3, COXI, and SDHA expression levels in female mice showed in (A). Error bars represent ± SEM. Statistical analysis was performed by one-way ANOVA followed by Bonferroni post-test.

C   Western blot of BN-PAGE of quadriceps of 6-month-old mice. We analyzed CI (NDUFA9, NDUFB8), CIV (COX1), and CII (SDHA).

D   Quantification of the BN-PAGE Western blots from female mice shown in panel (C). Error bars represent ± SEM. Statistical analysis was performed by one-way ANOVA followed by Bonferroni post-test.

E   BN-PAGE in-gel activity of CI and CIV in quadriceps of 6-month-old mice.

F   Quantification of BN-PAGE in-gel activities in females. Error bars represent ± SEM. Statistical analysis was performed by one-way ANOVA followed by Bonferroni post-test.

G   Total mtDNA levels in quadriceps from males and females were determined by qPCR using the ratio ND1/18S. Inset shows the appearance of the quadriceps, reflecting normalization of mitochondrial levels after replacement of NDUFS3. Error bars represent ± SEM. Statistical analysis was performed by one-way ANOVA followed by Bonferroni post-test.

H   Spectrophotometric CI/citrate synthase activity ratio was measured in quadriceps homogenates of 6-month-old males and females. Error bars represent ± SEM. Statistical analysis was performed by one-way ANOVA followed by Tukey post-test.

I   Plasma lactate levels were determined as described in methods. Error bars represent ± SEM. Statistical analysis was performed by one-way ANOVA followed by Tukey post-test.

mitochondrial proliferation in our smKO model could be specific to the CI defect. The absence of CI observed in our model may activate the signal transduction system employed in the coordination of the expression of nuclear and mitochondrial OXPHOS genes.

The *Ndufs3* smKO model recapitulated many features observed in patients with mitochondrial myopathies, such as elevated serum lactate levels and exercise intolerance. The characterization of this CI-deficient model showed that it was well suited for gene replacement therapy. Thus, we delivered rAAV9-*Ndufs3* systemically via retro-orbital injections in pre- or post-symptomatic groups of animals. We showed a complete prevention of the motor phenotype and restored NDUFS3 protein levels in the smKO mice injected with rAAV9-*Ndufs3* between 15–18 days old (pre-symptomatic group). The treatment also blunted the compensatory mitochondrial proliferation observed in the *Ndufs3* smKO mice. A single injection in 2-month-old symptomatic smKO mice already exhibiting clear signs of exercise intolerance and muscle weakness showed a marked recovery of motor phenotype 3–3.5 months after injection, with the KO-NDUFS3 behavior becoming essentially identical to unaffected mice. The remarkable rescue of NDUFS3 protein expression levels and activity correlates with the myopathy improvement. PGC-1α levels were also markedly reduced in the KO-NDUFS3 animals. Moreover, the total mtDNA levels and OXPHOS proteins analyzed were also essentially identical to unaffected mice. Therefore, the PGC-1α-mediated increase in mitochondrial biogenesis was reversible. Lactate levels were also normalized for both groups of injected animals, which is a reliable marker for mitochondrial myopathies.

Our results provide evidence for the remarkable ability of skeletal muscle to recover after weeks of being essentially without a functional OXPHOS. The increased number of myofibers with central nuclei suggests an active regenerating system, but we did not observe massive muscle degeneration. It is likely that most fibers can survive for extended periods of time relying on glycolysis/glycogen. However, these fibers cannot perform normal contractile functions. Upon NDUFS3 replacement, within weeks, myofibers restored normal function and reversed the abnormally increased mitobiogenesis.

Gene therapy strategies have emerged as possible treatments for mitochondrial-associated diseases. Torres-Torronteras *et al* (2011) successfully restored normal nucleoside pools in plasma, small intestine, skeletal muscle, brain, and liver of Tymp$^{-/-}$ mice, a model of MNGIE (mitochondrial neurogastrointestinal encephalopathy) disease, using lentiviral particles carrying a wild-type form of thymidine phosphorylase. Mitochondrial myopathy associated with muscle weakness and progressive external ophthalmoplegia (PEO) can be caused by mutations in heart–muscle isoform of ANT1. Recombinant AAV carrying the mouse Ant1 cDNA was used for direct injection in the gastrocnemius muscle of a mouse model KO for the muscle isoform of ANT1, resulting in improvements in the histopathology (Flierl *et al*, 2005). Barth syndrome (BTHS) is a rare mitochondrial disease that affects heart and skeletal muscle. It is caused by recessive mutations in the X-linked gene *TAZ*, which

**Figure 5. Replacement of NDUFS3 in 2-month-old (P60) smKO mice reverted the mitochondrial myopathy.**

A   Gene therapy strategy used in 2-month-old mice.

B   The smKO mice group that received retro-orbital injections at 2 months was tested prior to the rAAV9-*Ndufs3* injection showing a significant exercise intolerance attested by the number of falls in the treadmill. The KO-NDUFS3 gradually recovered over time. Individual KO-NDUFS3 mice are color coded. Green triangles represent the animals injected with GFP, whereas white triangles represent non-injected mice. Error bars represent ± SEM. Statistical analysis was performed by one-way ANOVA followed by Bonferroni post-test.

C   The hang test was performed in symptomatic mice, which had a significant decrease in four-limb muscle strength and coordination, at 2 months. The KO-NDUFS3 mice showed a significant recovery 3.5 months after injection. Individual KO-NDUFS3 mice are color coded. Green triangles represent the animals injected with GFP, whereas white triangles represent non-injected mice. Error bars represent ± SEM. Statistical analysis was performed by one-way ANOVA followed by Bonferroni post-test.

D   KO-NDUFS3 muscle weight was similar to WT^Flx mice, and it was significantly increased in gastrocnemius when compared to KO-GFP mice. Error bars represent ± SEM. Statistical analysis was performed by one-way ANOVA followed by Bonferroni post-test.

E   Immunohistochemical analysis with NDUFB8 and COX1 antibodies in quadriceps muscle sections of 6-month-old animals. The expression of NDUFB8 (representative of assembled CI) was recovered in the KO-NDUFS3 mice, and COX1 expression was similar to WT^Flx animals.

F   COX and COX/SDH activity staining showed that KO-NDUFS3 quadriceps were reverted to levels found in wild type, at 6 months.

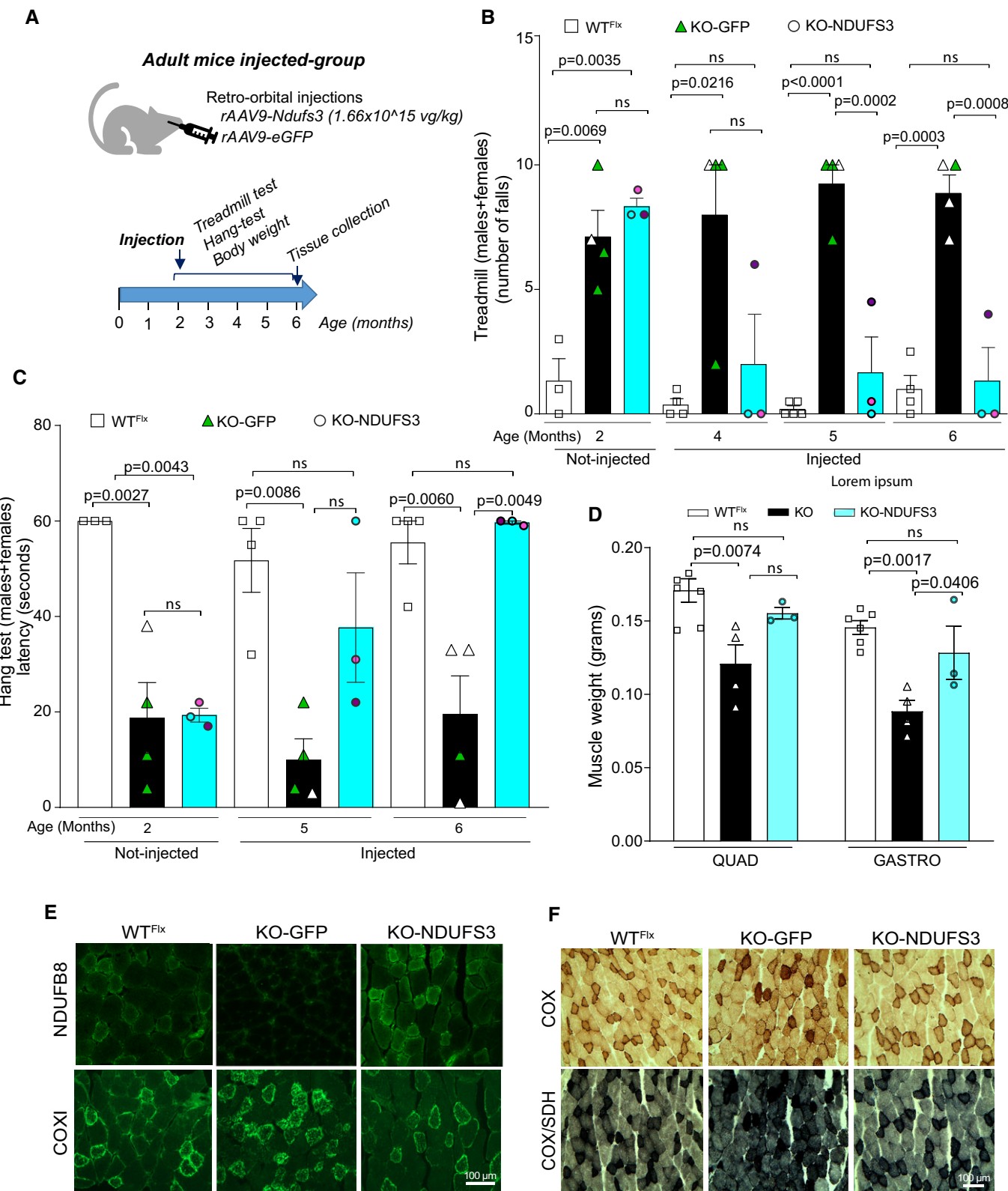

**Figure 5.**

encodes tafazzin. Recombinant rAAV9-TAZ showed a certain level of phenotypical improvement in a BTHS mouse model (Suzuki-Hatano *et al*, 2019).

Gene therapy was also recently applied in *Ndufs4*-KO newborn and young mice, using rAAV2/9-hNDUFS4, which partially ameliorated the mice clinical phenotype (Di Meo *et al*, 2017). However,

the mild improvement was only possible with the combination of intravenous (IV) and intracerebroventricular (ICV) administration routes, and it could only partially ameliorate the mice motor phenotype and extend survival for a short period of time (Di Meo et al, 2017). Clinical trials are currently ongoing for LHON, using rAAV-mediated allotopic expression of ND4, which consists in the

expression of a normally mtDNA-coded gene in the nucleus (Garone & Viscomi, 2018).

In conclusion, different studies have now shown that rAAVs can efficiently infect both dividing and nondividing muscle cells (Pruchnic et al, 2000) and allow long-term expression of the genes (Xiao et al, 1996; Fisher et al, 1997). However, the ability

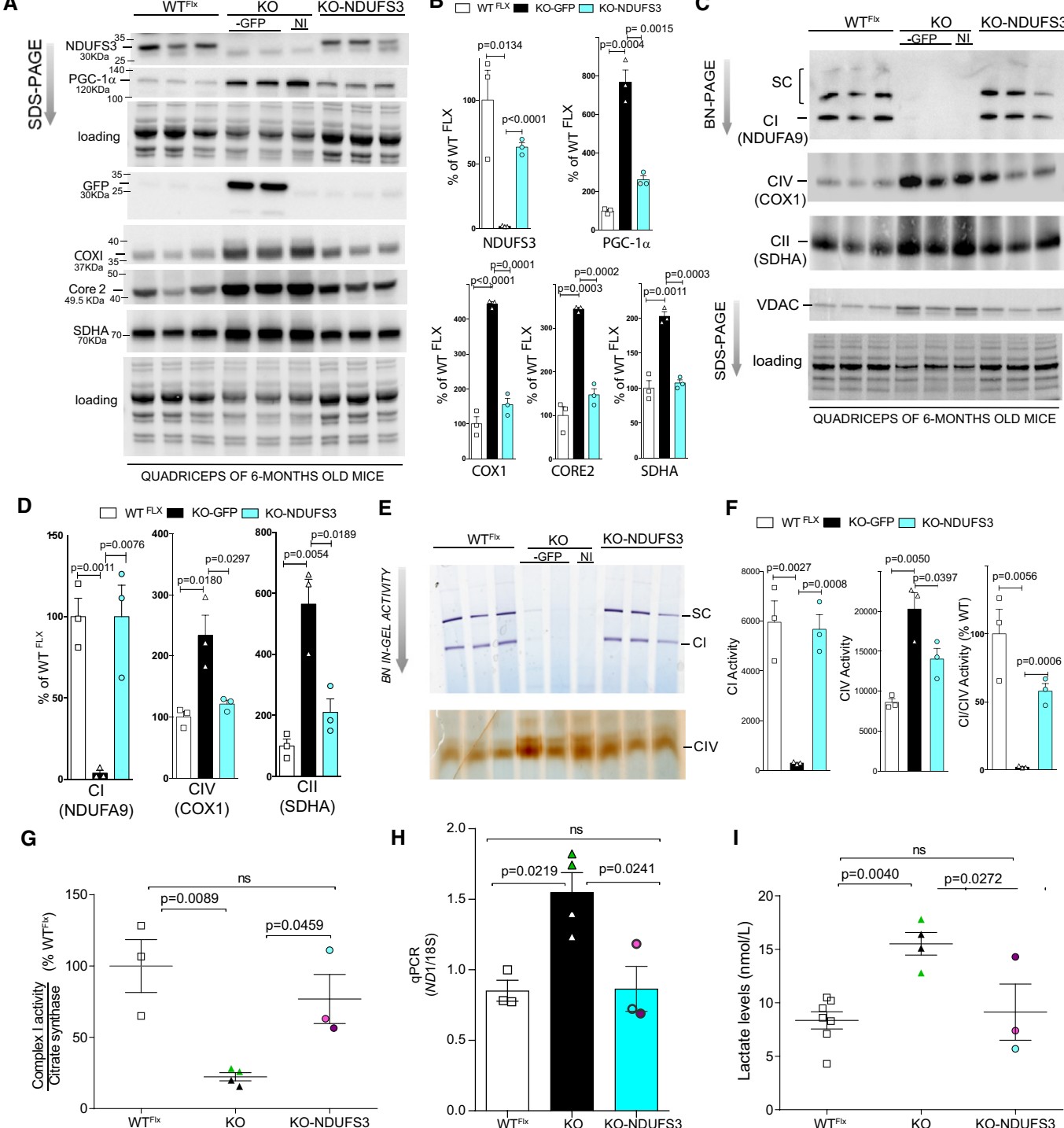

Figure 6.

**Figure 6.  NDUFS3 replacement (at P60) normalized the levels of lactate and mitochondrial markers.**

A   Western blots showed NDUFS3 expression in KO-NDUFS3 and GFP expression in the KO-GFP quadriceps samples. PGC-1α expression was normalized in the KO-NDUFS3 mice, which was abnormally elevated in the KO-GFP. The same was observed for CIV (COX1), CIII (Core 2), and CII (SDHA).

B   Quantification of Western blots shown in (A). Error bars represent ± SEM. Statistical analysis was performed by one-way ANOVA followed by Bonferroni post-test.

C   BN-PAGE Western blots showed recovery of complex I assembly into complex and supercomplexes in KO-NDUFS3-injected mice. CIV and CII were similar to wild type in the KO-NDUFS3 samples, at 6 months.

D   Quantification of BN-PAGE Western blots shown in (B). Error bars represent ± SEM. Statistical analysis was performed by one-way ANOVA followed by Bonferroni post-test.

E   BN-PAGE in-gel CI and IV activities of KO-NDUFS3 mice were also reverted to levels similar to the WT$^{Flx}$ samples.

F   Quantification of in-gel activities. Error bars represent ± SEM. Statistical analysis was performed by one-way ANOVA followed by Bonferroni post-test.

G   Complex I/citrate synthase activity of KO-NDUFS3 samples measured spectrophotometrically were similar to WT$^{Flx}$ samples and increased in comparison with KO-GFP mice. The individual KO-NDUFS3 animals are color coded. Green triangles represent the animals injected with rAAV9-*eGFP*, whereas black triangles represent animals that did not receive an rAAV9-*eGFP* injection. Error bars represent ± SEM. Statistical analysis was performed by one-way ANOVA followed by Tukey post-test.

H   Total mtDNA levels were determined by qPCR. MtDNA levels in KO-NDUFS3 mice were decreased when compared to KO-GFP mice and not different from the levels in WT$^{Flx}$ mice. The individual KO-NDUFS3 animals are color coded. Green triangles represent the animals injected with rAAV9-*eGFP*, whereas black triangles represent animals that did not receive an rAAV9-*eGFP* injection. Error bars represent ± SEM. Statistical analysis was performed by one-way ANOVA followed by Bonferroni post-test.

I   The plasma lactate levels were similar to WT$^{Flx}$ in KO-NDUFS3 samples and significantly decreased when compared to KO-GFP levels. The individual KO-NDUFS3 animals are color coded. Green triangles represent the animals injected with rAAV9-*eGFP*, whereas black triangles represent animals that did not receive an rAAV9-*eGFP* injection. Statistical analysis was performed by one-way ANOVA followed by Tukey post-test. Data represent means ± SEM.

of completely restoring function of essentially all muscles after prolonged OXPHOS defect was an interesting observation. Realistically, patients with an OXPHOS defect in muscle are diagnosed when the pathology and symptoms are already present. Our results showed that gene replacement therapy could restore normal muscle function even after a severe mitochondrial myopathy has developed and compensatory mechanisms (e.g., mitochondrial proliferation) have been established. A single high titer injection of rAAV9-*Ndufs3* was enough to replace the missing gene in all mouse skeletal muscles analyzed, and expression appears to last for at least 15 months (the oldest mice analyzed).

Although this welcome feature reflects the plastic physiological properties of skeletal muscle, at this point it is not known whether the same applies to other tissues, such as CNS. In any case, if we extrapolate from mice to human, our study suggests that gene replacement can be applied to a relatively wide temporal window when treating mitochondrial myopathies.

## Materials and Methods

### Creation of skeletal muscle-specific Ndufs3 conditional KO mice

We acquired the floxed *Ndufs3* mice from the Knockout Mouse Project repository [(https://www.komp.org/geneinfo.php?geneid = 70352, accessed on 22 February 2019), Appendix Fig S1]. To generate the *Ndufs3* muscle-specific knockouts, males homozygous for a floxed *Ndufs3* (*Ndufs3*$^{f/f}$) were mated to double heterozygous females *Ndufs3*$^{(+/f)}$, Mlc1f-cre$^{(+/-)}$, and the resulting progeny with the desired genotype selected for phenotypic studies. All animals used were males and had a pure C57Bl/6J background, backcrossed for at least 10 generations. All experiments and animal husbandry were performed according to a protocol approved by the University of Miami Institutional Animal Care and Use Committee. Mice were housed in a virus/antigen-free facility of the University of Miami, Division of Veterinary Resources in a 12-h light/dark cycle at room temperature and fed *ad libitum*. Conditional KO mice were injected with either AAV-GFP or AAV-NDUFS3 viral preparations (randomized).

### Behavioral tests

#### Ambulatory activity

Spontaneous ambulatory movement of mice was recorded using the Opto-M3 activity meter (Columbus Instruments) equipped with infrared beams along the cage. Movement was counted as the number of times the infrared beams were disrupted. Mice were housed individually in a new cage 30 min prior to their daily dark cycle, and ambulatory counts were recorded for a period of 12.5 h (6:30 pm to 7 am).

#### Open field

Open field (Med Associates Inc.) is a sensitive method for measuring gross and fine locomotor activity. It consists of a chamber and a system of 16 infrared transmitters that record the position of the animal in the three-dimensional space. With this system, not only the horizontal movement can be recorded but also the rearing activity. For our study, the animals were placed in the chamber 30 min before the test and the locomotor activities were recorded for 30 min.

#### Treadmill

Mice motor coordination was tested at different ages using a Treadmill (Columbus instrument Exer 3/6) set at a speed of 7.5 m/min. The mice were running for 3 min, and the number of falls in the grid was recorded. If the animals could not perform, a score of 10 was given. Animals were trained in the treadmill 1 week before the test.

#### Rotarod test

Mice motor coordination was tested at different ages using a Rotarod (IITC 755 Life Sciences) set at a ramp speed of 6–20 rpm over 180 s. The test consisted of three trials performed for each animal at the corresponding age and the latency to fall was recorded. Mice that completed the task received a final latency time of 180 s. Animals were trained in the rotarod two times of three trials each about 2 weeks prior to the first test.

### Hang test

The four-limb hang test consisted in placing the mice in a metal grid and count the time they remained on it, opposing their gravitational force. The mice were given 60 s to remain on the grid. If the mice fell off before the time limit, they were given two more tries. The best time was considered for statistical analysis. Soft bedding was placed underneath the grid to break the fall and prevent the mice from harming themselves. The grid was placed high enough to reduce the tendency of the mice to jump. The mice were evaluated at different ages, and a training was performed prior to the real test.

### Vector construction, production, and injection

The RNA from lung mouse tissue was isolated as described (RNeasy Mini Kit, Quiagen). The RNA was converted to cDNA by reverse transcriptase reaction (SuperScript™ III First-Strand Synthesis System, Invitrogen). The appropriate primers (F-ACCGTT TAAACT CGAGTAGTCGGCACACTAAGGAA; R-CCGCGATATCGGATCCTTAC TTGGT T TCAGGCTT) were designed to amplify the *Ndufs3* cDNA from lung tissue of a C57BL/6J wild-type mouse. The PCR product (844 bp) was separated and purified from an agarose gel, as described (NucleoSpin Gel extraction kit, Macherey-Nagel). Next, we have used an AAV vector digested with XhoI and BamHI in which the PCR fragment containing *Ndufs3* cDNA was cloned by Infusion (In-Fusion® HD Cloning, Clontech). The final construct was sequenced verified and sent for viral production of rAAV9 at the University of Iowa Carver College of Medicine Viral Vector core. For systemic delivery of rAAV9-*Ndufs3*, mice at P15–18 or at P60 were subjected to a single retro-orbital injection with $1.25 \times 10^{15}$ vg/kg (in P15–18) or $1.66 \times 10^{15}$ vg/kg (in adults). All injections were carried out using a short insulin syringe with a 31G needle (Becton Dickenson). Tail tissue was obtained at P21 for the adult mice and finger tissue at P3 for the P15- to 18-day-old mice, before performing the injections to determine their genotype. The mice were re-genotyped after sacrifice (at 6 months), by collecting a small tail tissue sample.

### Western blots

Protein extracts were prepared from skeletal muscle and homogenized in 1× PBS containing a protease inhibitor mixture (Roche). Upon use, sodium dodecyl sulfate (SDS) was added to the homogenate at the final concentration of 4%. Homogenates were then centrifuged at 14,000 *g* at 4°C, and the supernatant was collected for analysis. Protein concentration was determined by Lowry assay using the BCA kit (Bio-Rad). Protein (20 µg) was separated by SDS–PAGE in 4–20% acrylamide gels and transferred to PVDF membranes (Bio-Rad). Membranes were blocked with 5% non-fat milk in 0.1% Tween-20 in PBS and subsequently incubated with specific antibodies, which were incubated overnight at 4°C. Antibodies against NDUFS3, NDUFB8, SDHA, COX1, TIM23, Core2 (UQCR2), and VDAC1 were obtained from Abcam, diluted 1:1,000; ACTIN and TUBULIN from Sigma, 1:5,000; and PGC-1α from Proteintech 1:1,000. Secondary antibodies conjugated to horseradish peroxidase (Cell signaling) were used, and the reaction was developed by chemiluminescence using SuperSignal West reagent (Thermo Fisher, Rockford, IL). Blots were visualized with Chemi-

doc Imaging System (Bio-Rad). Optical density measurements were taken by software supplied by Bio-Rad (Image Lab). The use of TGX Stain-Free™ Precast Gels allowed the determination of total protein loading through gel activation with the imager, following manufacturer's instructions.

### Enzymatic activity assays

Muscle homogenates were prepared in PBS containing complete protease inhibitor cocktail (Roche diagnostics) in a volume of 10× the weight. The tissues were disrupted by 10–15 strokes, using a motor-driven pestle. Homogenates were centrifuged at 1,000 *g* for 5 min and supernatants used for enzymatic assays. The activities of CI and citrate synthase were measured spectrophotometrically as described previously (Barrientos, 2002; Spinazzi *et al*, 2012). Protein concentrations were determined using the Bio-Rad Bradford Assay Kit with bovine serum albumin (BSA) as standard. Specific activity was quantified as % of the wild-type mice.

### Blue native PAGE

To identify and estimate the levels of respiratory supercomplexes, homogenates from skeletal muscle were treated with digitonin (ratio 1:8, protein: digitonin, Roche) and mitochondrial complexes separated by blue native PAGE (BN-PAGE) in 3–12% acrylamide gradient gels (Invitrogen; Wittig *et al*, 2006; Diaz *et al*, 2009). Protein (10–20 µg) was used and separated by PAGE, transferred to a PVDF membrane (Bio-Rad), and incubated sequentially with antibodies against several subunits of the different mitochondrial respiratory chain complexes. To detect the activity of complex I in-gel, samples (60 µg) treated with digitonin were separated in 3–12% gels and incubated with 0.14 mM NADH and 1 mg/ml of nitroblue tetrazolium in Tris–HCl 0.1 M pH 7.4. The gels were incubated at 37°C for approximately 1 h. To detect the activity of complex IV, the gels were incubated with 5 mg of diaminobenzidine (DAB) dissolved in 9 ml of 50 M phosphate buffer pH 7.4 containing 10 mg cytochrome *c* and 750 mg of sucrose. The gels were incubated at 37°C until a brown color was observed.

### Histology

Skeletal muscles were frozen in isopentane cooled by liquid nitrogen. Cryostat sections of 10 µm were stained with hematoxylin–eosin to study muscle structure. Cytochrome *c* oxidase (COX) activity stain and succinate dehydrogenase (SDH) activity stain were performed as described (Peralta *et al*, 2016). Briefly, for COX staining, sections were incubated in 100 mM sodium phosphate buffer pH 7.4 containing 0.5 mg/ml diaminobenzidine, 0.2 mg/ml cytochrome *c*, and 40 mg/ml sucrose at 37°C for 50 min. For SDH activity, stain sections were incubated in 10 mM sodium phosphate buffer pH 7.5 containing 1.6 mg/ml EDTA, 0.65 mg/ml KCN, 0.06 mg/ml phenazine methosulfate, 1.3 mg/ml succinic acid, and 1.2 mg/ml nitroblue tetrazolium at 37°C, for 20 min. For double COX/SDH stain, sections were first stained with COX reagents, washed three times in PBS, and then incubated with SDH reagents. Images were captured with an optic microscope. For immunofluorescent staining, sections were blocked with 10% normal goat serum (NGS) for 1 h at RT and permeabilized with 1% Triton X-100. Sections were incubated with primary Ab, NDUFS3 1:500 (Abcam) or NDUFB8 1:200 (Abcam); anti COX1 1:500 or 1:1,000 (Abcam), overnight at 4°C. Slides were then incubated with Alexa

**The paper explained**

**Problem**

Mitochondrial myopathies can be caused by mutations in either the mitochondrial or the nuclear DNA. Patients suffering from mitochondrial myopathies have a defect in one of the five oxidative phosphorylation system. Complex I is the largest complex with 45 subunits in humans.

**Results**

We generated a new model of mitochondrial myopathy by the conditional deletion of a complex I subunit (NDUFS3) in skeletal muscle. The mice showed a severe myopathy, starting at approximately 2 months of age. We replaced the missing gene in muscle by systemic injection of recombinant AAV9-expressing *Ndufs3*. Injection of young mice (approximately 17 days old) prevented the development of the myopathy while injection of myopathic adult mice (60 days old) reverted the phenotype.

**Impact**

These results showed that skeletal muscle with a severe OXPHOS defect can be rescued and function can be restored if the genetic defect is corrected. These findings indicate that muscle is not permanently impaired by a prolonged mitochondrial complex I defect.

Fluor secondary antibodies for 1 h at RT and mounted with Vectashield containing DAPI mounting medium for fluorescence microscopy. Images were captured with an Olympus BX51 confocal microscope and/or with a fluorescence microscope (Invitrogen™ EVOS™ Digital Color).

## Transmission electron microscopy

Quadriceps muscles from *Ndufs3 Mlc1f*-KO and wild type of 8 months old were fixed in 2.5% glutaraldehyde overnight and treated with 1% osmium tetroxide, dehydrated in ethanol, and embedded in epoxy resin. Semi-thin (1 μm) sections were stained with Richardson's stain. For transmission electron microscopy (TEM), ultra-thin sections (100 nm) were cut on grids, stained with uranyl acetate, and lead citrate and scope in a Philips CM-10 transmission electron microscope equipped with a Gatan Erlangshen ES 1000W digital camera.

## Plasma lactate measurements

Blood was withdrawn from deeply anesthetized animals by cardiac puncture. Lactate levels were measured in plasma using Lactate Plus Lactate Meter (Nova Biomedical).

## Quantitative PCR of genomic DNA

Genomic DNA was extracted from skeletal muscle tissue using standard proteinase K, phenol, chloroform extraction, and isopropyl alcohol precipitation. The qPCR was performed with PrimeTime qPCR probes (IDT, Integrated DNA Technologies) according to the manufacturers' protocol (IDT) for both mtDNA (ND1) and genomic DNA (18S). The qPCR was performed on a Bio-Rad CFX96/C1000 qPCR machine. We used the comparative cycle threshold ($C_t$)

method to determine the relative quantity of mtDNA. The level of mtDNA was determined by quantifying the levels of total mtDNA/genomic DNA (ND1/18S). The following primers/probes were used: ND1- F: GCC TGA CCC ATA GCC ATA AT (NC_005089; mtDNA 3282-3301); ND1-R-CGG CTG CGT ATT CTA CGT TA (mtDNA 3402-3383). Probe:/56-FAM/TCT CAA CCC/ZEN/TAG CAG AAA CAA CCG G/3IABkFQ/(mtDNA 3310-3334), for mtDNA. For genomic DNA: 18S-F: GCC GCT AGA GGT GAA ATT CT (RefSeq NR_046233.2; chr17:39984253-39984272); 18S-R-TCG GAA CTA CGA CGG TAT CT (RefSeq NR_046233.2; chr17:39984432-39984412). Probe:/5Cy5/AAG ACG GAC CAG AGC GAA AGC AT/3IAbRQSp/(RefSeq NR_046233.2; chr17:39984285-39984305).

## Viral genome determination by quantitative PCR

The qPCR was performed on a Bio-Rad CFX96/C1000 machine, and genomic DNA was extracted as previously described. PrimeTime qPCR probes (IDT, Integrated DNA Technologies) were directed to the NDUFS3 insert, one including part of the CMV promoter and the other including part of the bGH insert. The PCR cycle conditions used were the following: 95°C for 3 min, 95°C for 15 s, and 60°C for 1 min, 39 cycles. Genomic DNA (125 ng) was diluted in a final reaction volume of 10 μl. Standard curve was determined using $10^1$–$10^9$ copies of the AAV-NDUFS3 plasmid used to make the virus. All qPCR products were performed together with the standard curve, and samples were run in triplicate. Viral genome copies of injected animals were calculated subtracting the vg of injected mice samples with AAV9-NDUFS3 from non-injected mice samples. At least three independent experiments were performed, and vg copies were express by 100 ng of DNA for each tissue. The following primers/probes were used:
CMV-NDUFS3: Probe: 5′-TET/AGCAGAGCT/ZEN/CGTTTAGTGAACCGT-3′
F: 5′-TCCTTAGTGTGCCGACTAACT-3′, R: 5′-GTGTACGGTGGGAGGTCTAT-3′;
NDUFS3-bGH Probe: 5′-FAM/AAGTAAGGA/ZEN/TCCGATATCGCGGCC-3′;
F: 5′-CAGATGGCTGGCAACTAGAA-3′, R: 5′-CTCGAAGCTGGAGACAAGAAG-3′.

The number of DNA copies was determined by the following equation: (amount of dsDNA in nanograms × (6.022 × $10^{23}$))/(length of ss DNA in base pairs × (1 × $10^9$*330)).

## Statistical analysis

Two-tailed, unpaired Student's *t*-test was used to determine the statistical significance between two different groups. Multiple groups were compared using a one-way ANOVA followed by a Tukey or Bonferroni *post hoc* comparisons, as indicated in figure legends. Conditional KO mice were injected (randomized) with rAAV-GFP or rAAV-*Ndufs3*. The number of animals used is indicated in the graph or in figure legends. Error bars represent ± SEM. *P* values are indicated in the graphs or represented by asterisks, *$P < 0.05$ was considered significant. GraphPad prism 8 software was used for the presentation of the data.

**Expanded View** for this article is available online.

## Acknowledgements

We are grateful to Michelle Barragan, Alexis Lopez, Aline Hida, and Paula Lima for assistance with mice genotyping and behavior. This work was partially supported by the National Institutes of Health grants (R01EY0108041, R01AG036871, R01NS079965, R33ES025673), the CHAMP Foundation, and the Muscular Dystrophy Association.

## Author contributions

CVP and SP conducted most of the experiments and wrote the manuscript. TA performed mice behavioral and Western blots; SRB assisted with the retro-orbital injections, total mtDNA levels, and viral genome copies determination. FD assisted with BN-PAGE experiments. CTM planned the project together with CVP and SP and contributed to the writing of the manuscript. All authors edited the manuscript.

## Conflict of interest

The authors declare that they have no conflict of interest.

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
