## [Review Process File · EMBO Molecular Medicine]

Myopathy reversion in mice after restauration of mitochondrial complex I

Claudia V. Pereira, Susana Peralta, Tania Arguello, Sandra R. Bacman, Francisca Diaz and Carlos T. Moraes

Review timeline:

Submission date:	27 March 2019
Editorial Decision:	25 April 2019
Revision received:	5 November 2019
Editorial Decision:	19 November 2019
Revision received:	4 December 2019
Accepted:	6 December 2019

Editor: Céline Carret

Transaction Report:

1st Editorial Decision

25 April 2019

Thank you for the submission of your manuscript to EMBO Molecular Medicine. We have now heard back from the three referees whom we asked to evaluate your manuscript.

You will see from the set of comments pasted below that while ref. 1 and 2 are supportive of publication, ref. 3 is more critical and highlights a few limitations of the study. Upon our cross-commenting exercise, it appears that indeed analyses of animals should be repeated by separating males from females, n increased when too low, mitochondrial fragmentation, O₂ consumption rates and vector genome distribution analyzed, and results quantified when suggested while missing details and explanations must be provided.

We would therefore welcome the submission of a revised version within three months for further consideration and would like to encourage you to address all the criticisms raised as suggested to improve conclusiveness and clarity. Please note that EMBO Molecular Medicine strongly supports a single round of revision and that, as acceptance or rejection of the manuscript will depend on another round of review, your responses should be as complete as possible.

I look forward to receiving your revised manuscript.

***** Reviewer's comments *****

Referee #1 (Comments on Novelty/Model System for Author):

The paper shows that a single injection of an AAV vector containing the WT version of a missing but essential gene encoding a protein of complex I is able to prevent and even restore the normal phenotype in a muscle-restricted recombinant KO mouse.

Referee #1 (Remarks for Author):

This is a beautiful and interesting paper showing the efficacy of gene replacement therapy at least in a non-neurological mitochondrial experimental model. The experimental work is sound and very clear and the conclusions sound and straightforward.

Referee #2 (Remarks for Author):

Thank you very much for sending me this manuscript by Pereira et al. to review. It concerns the production of a skeletal muscle-specific knockout of the mitochondrial respiratory complex I subunit, NDUFS3. The ts KO mice is well described and characterised, showing a profound loss of fully assembled complex I (and associated supercomplexes) and a profound mitochondrial proliferation potentially as an attempt to compensate for this loss of OXPHOS. This is remarkable and I have never seen such an example of mito proliferation. This work alone is important, as the field does not yet have a good number of models of OXPHOS deficiency. There is a claim of sarcopenia, but looking at the sections in Fig 5 (please add size bar to 5F) there is little evidence of this after 6 months. Deficiency is rescued on injection of an AAV expressing wt NDUFS3 at P15, as judged by numerous parameters. What is really impressive and eye-catching about this work is that the muscle-specific defects and increased serum lactate that are apparent in the KO mouse after 2 months can be reversed by systemic injection of the AAV-NDUFS3. This is a key observation, as it infers that it may be possible to restore muscle function to patients who have similar complex I defects post clinical presentation.

Overall, the expts and analyses in this paper are performed expertly and I have little to question. I therefore believe this paper is suitable for publication in EMBO Mol Medicine. My only slight concern is that nuclear encoded defects of complex I components often cause major brain disorders, such as Leighs Disease as discussed by the authors. It would be very interesting to produce a similar ts mouse affected in, perhaps, neocortex/hippocampus and whether this can also be rescued by the AAV, particularly as the authors suggest they were able to find evidence of AAV in the brain. However, this presupposes that the ts KO would not be lethal.

Referee #3 (Comments on Novelty/Model System for Author):

Technical quality is low because there are low sample sizes in some of the data presented in this manuscript. Novelty is low because it is a straight-forward provide a gene in a KO mouse story that is not thoroughly done from a gene delivery perspective.

Referee #3 (Remarks for Author):

This manuscript describes a study that characterizes the mitochondrial phenotype of a muscle tissue specific mouse model for mitochondrial complex I deficiency and demonstrates reversal of the muscle pathology in mice treated with gene replacement. While this is a good model of mitochondrial myopathy and the gene delivery aspect demonstrates feasibility for this family of disorders, it is not a novel concept to replace the deficient gene in a knockout model of a mitochondrial disorder so although it is a decent gene delivery study the true novelty of this is unclear in its present state.

Specific comments:

1. Introduction, last paragraph - The authors conclude that the study implies that a wide temporal therapeutic window for gene therapy is possible for mitochondrial myopathies when the 6 weeks difference in age presented in this study is not as vast a window as demonstrated in many other contemporary gene therapy studies including some involving mitochondrial disorders.

2. Figure 1 - As there is a difference in the age of onset based upon significant weight differences for males and females in this model, please present male and female Ambulatory, treadmill, and rotarod data separately in the initial characterizations so that subsequent treated groups can be properly evaluated.
3. Figure 1 - The animal numbers in some groups appear to be very low (n=2???).
4. Figure S2A-B - Please define stereotypical time and activity. The tracings appear to show significant differences between mice in the same group - please explain.
5. Figure 2 - The authors state that the muscle fibers appear smaller in ndufs3 mice, as you have the images, please quantify this.
6. Figure 2 - n = 3 sections? What does this indicate - three sections from each of 3 mice or 3 sections from the same mouse - these are not robust sample size numbers for this type of data set.
7. Figure 2E - the Complex IV data is not clean. Please quantify all of these to determine significance.
8. Figure S3 - the authors conclude there is increased mitochondrial proliferation but it is important to evaluate these samples for markers of mitochondrial fragmentation as the EM images suggest this may also be occurring. EM data would be stronger if quantified.
9. Figure 3 - Please indicate what dose was used in vg/kg for the different treatment times.
10. Figure 4 - The authors describe the treatment as showing restored NDUF3 protein levels in all muscle homogenates analyzed. Please show these data for other muscles and quantify as done earlier in the manuscript. Why does one male appear to express far more GFP than the others and why does there appear to be more NDUF3 expression in treated females than males?
11. Figure 4 - Why are only females included in 4B and C? Please include both males and females and quantify the data. D-F are these males or females? - please separate and quantify.
12. Figure S4 and S5 - please switch these as they are called out 5 then 4 in the text.
13. Figure S5 - this does not represent AAV9 distribution. Please present vg data for tissues to show AAV9 distribution.
14. Figure S5 - Why does panel A show a combination of 6 and 15 month data and B is only 6 month data? Please provide replicates, quantify all of these data, and present in a similar format across the injection times for all information in S5.
15. Figure 5 - Please separate these data between males and females and provide an explanation for what has changed between 5 and 5.5 months for 5C. Again - sample sizes are low for these types of data.
16. Figure 5D - Please separate males and females.
17. Figure 5F - please quantify these data across multiple sections from multiple mice - the NDUF3 treated group looks to have fewer positive fibers than WT... please explain. Is there evidence of fiber type switching in this model with or without treatment?
18. Figure 6A, B, C - [please quantify results].
19. Methods - Vector description needs to include an actual dose/kg for injections.
20. Discussion - Please discuss cause of death in this model.
21. As the authors list only 2 human patients with some form of NDUF3 disorder and mention other groups who have shown gene therapy type improvements in other mitochondrial genetic disorders, please provide more discussion on how this study is innovative.
22. Discussion page 2 - the authors say the 2 month injections are post-symptomatic but that may not be true for females based on the data as presented - when male/female data are separated out it will be potentially possible to draw these conclusions.
23. Discussion page 3 - The authors state that AAV is non-pathogenic but this is clearly a hotly debated topic in the field. Please address these issues and other limitations.

General Comments:

1. It is important to include vg and RNA transcript data in addition to protein data in studies such as this so that mechanistic conclusions regarding the level of regulation for these interacting proteins can be drawn.
2. Mitochondrial proliferation is mentioned several times but often mitochondrial fragmentation is a huge issue for muscle related mitochondrial disorders so evaluating these markers will provide a more complete picture of the impact of this deficiency.
3. All the characterizations performed in the initial figures should be repeated in the treated groups - EM, 2 month treated body weights for male and female curves over time and centralized nuclei measurements.
4. This is a mitochondrial story so providing oxygen consumption data would greatly enhance your study by providing direct mitochondrial functional data...all mitochondrial researchers reading this

in its current state would be left disappointed and wondering what oxygen consumption would look like in these animals.

1st Revision - authors' response

5 November 2019

Referee #1 (Remarks for Author)

The paper shows that a single injection of an AAV vector containing the WT version of a missing but essential gene encoding a protein of complex I is able to prevent and even restore the normal phenotype in a muscle-restricted recombinant KO mouse.

This is a beautiful and interesting paper showing the efficacy of gene replacement therapy at least in a non-neurological mitochondrial experimental model. The experimental work is sound and very clear and the conclusions sound and straightforward.

Thank you for the positive comments.

Referee #2 (Remarks for Author):

Thank you very much for sending me this manuscript by Pereira et al. to review. It concerns the production of a skeletal muscle-specific knockout of the mitochondrial respiratory complex I subunit, NDUFS3. The ts KO mice is well described and characterised, showing a profound loss of fully assembled complex I (and associated supercomplexes) and a profound mitochondrial proliferation potentially as an attempt to compensate for this loss of OXPHOS. This is remarkable and I have never seen such an example of mito proliferation. This work alone is important, as the field does not yet have a good number of models of OXPHOS deficiency. [1] There is a claim of sarcopenia, but looking at the sections in Fig 5 (please add size bar to 5F) there is little evidence of this after 6 months. Deficiency is rescued on injection of an AAV expressing wt NDUFS3 at P15, as judged by numerous parameters. What is really impressive and eye-catching about this work is that the muscle-specific defects and increased serum lactate that are apparent in the KO mouse after 2 months can be reversed by systemic injection of the AAV-NDUFS3. This is a key observation, as it infers that it may be possible to restore muscle function to patients who have similar complex I defects post clinical presentation.

Overall, the expts and analyses in this paper are performed expertly and I have little to question. I therefore believe this paper is suitable for publication in EMBO Mol Medicine. My only slight concern is that nuclear encoded defects of complex I components often cause major brain disorders, such as Leighs Disease as discussed by the authors. [2] It would be very interesting to produce a similar ts mouse affected in, perhaps, neocortex/hippocampus and whether this can also be rescued by the AAV, particularly as the authors suggest they were able to find evidence of AAV in the brain. However, this presupposes that the ts KO would not be lethal.

1- We have added size bars and have also changed the panels to newer ones that show additional fibers in Figure 5 (to also address Reviewer 3 comments). Although there was a clear reduction in muscle mass (please see Figure 2A, Figure 3H, Figure 5D, and Figure Supplementary S2D), we removed the term Sarcopenia from the manuscript.

2- We have developed a Ndufs3 specific KO in forebrain neurons, by using a Cre recombinase under the calcium/calmodulin-dependent protein kinase II (CaMKII α) promoter, which is expressed predominantly in cortex and hippocampus. This mouse model develops a severe and progressive encephalopathy and does not survive longer than 4 months. We are planning to test the efficiency of the AAV in brain. However, this will not only entail a whole new set of experiments and viral isotopes, which will take a couple of years to perform, but also it would tackle a different problem, which is a neuropathy. Therefore, it is beyond the scope of this study.

Referee #3 (Remarks for Author):

Reviewer 3 found the work interesting but had several criticisms:
The reviewer mentioned that the study lacks in novelty.

We have to disagree with this comment. We have created a novel model of a muscle complex I deficiency. To our knowledge there is no mouse model with an isolated complex I deficiency in muscle in the literature. The strong mitochondrial proliferation associated with the complex I defect in muscle is also novel. In addition, we are not only showing that we can replace NDUF3 in all muscle groups by gene therapy, at an age where the protein is already missing (15-18 days), but also after an overt myopathy has ensued (60 days). In the latter case, the phenotype was reverted. As noted by Reviewers 1 and 2, it is an exciting and novel development. In terms of gene replacement, there is one publication on a *Ndufs4* model, but it is a different study, as the mouse had mostly a CNS involvement and the functional recovery was very limited (DiMeo et al., 2017). In addition, there is an ANT model, which is a myopathy and there was a recovery, but it was not an OXPHOS complex and the AAV-ANT was injected intramuscular, not systemically (Flierl et al. 2005).

Specific comments:

1. Introduction, last paragraph - The authors conclude that the study implies that a wide temporal therapeutic window for gene therapy is possible for mitochondrial myopathies when the 6 weeks difference in age presented in this study is not as vast a window as demonstrated in many other contemporary gene therapy studies including some involving mitochondrial disorders.

The main point we tried to make was that the myopathy was reverted after the NDUF3 protein was missing for approximately 50-60 days, which corresponds to approximately 4-5 years in humans. The exact number is not critical, but the fact that the mitochondrial myopathy was reverted is a novel and important finding. We are not aware of similar results in the literature.

2. Figure 1 - As there is a difference in the age of onset based upon significant weight differences for males and females in this model, please present male and female Ambulatory, treadmill, and rotarod data separately in the initial characterizations so that subsequent treated groups can be properly evaluated.

We have used only males for the original extensive phenotypic characterization (Figure 1E-G and Figure Supplemental S2A-B). However, regarding the behavioral tests used for the gene therapy studies, we have also compared males and females at 2 months (Figure Supplemental S4A-B). These latter comparisons showed no significant difference in the onset of a myopathy between males and females (Figure Supplemental S4A-B). These were evaluated by the number of falls in the treadmill and by the latency to fall in the hang-test

3. Figure 1 - The animal numbers in some groups appear to be very low (n=2??).

The endurance test had two age groups, one of which had only 2 animals. We agree with the reviewer that this is too low and have eliminated this panel from Figure 1.

4. Figure S2A-B - Please define stereotypical time and activity. The tracings appear to show significant differences between mice in the same group please explain.

The panels of figure S2A showed the path of each individual mouse during the open field test. In a normal rodent exploratory behavior, the mice explore the field covering first the corners and after that, they explore the central part of the square. The paths suggest that *Ndufs3* smKO mice travel less distance for the same amount of time than their control littermates and in addition smKO mice do not cover the central region of the field as well as the control mice.

In the Figure S2B, the two graphs represent 2 ways of measuring similar parameters: stereotypical time and stereotypical counts. To simplify, we have eliminated the graph of stereotypical counts and left only the stereotypical time. Stereotype refers to repetitive behaviors in animals. The stereotypical time refers to the time the mice spend in grooming and sniffing behaviors in the open field test (total time of the test is 30 min).

The variability observed within each group of mice is not uncommon in behavioral tests.

5. Figure 2 - The authors state that the muscle fibers appear smaller in *ndufs3* mice, as you have the images, please quantify this.

We have now quantified the cross-sectional area (CSA) of the muscle fibers of quadriceps and included it in figure 2, panel C. As expected, the cross-sectional area of the muscle fibers of *Ndufs3* smKO mice was smaller.

6. Figure 2 - n = 3 sections? What does this indicate - three sections from each of 3 mice or 3 sections from the same mouse - these are not robust sample size numbers for this type of data set.

We used n=3 mice for controls and n=4 mice for smKO.

7. Figure 2E - the Complex IV data is not clean. Please quantify all of these to determine significance.

The gel depicted in Fig. 2E showed only 2-3 samples of wild-type and mutant mice at 2 and 4 months. The in-gel activity illustrated an essentially complete loss of complex I and an apparent increase in complex IV. Because of the low “n” in this gel we did not show the quantification. These data were expanded later for 6-month mice with a higher n and statistical treatments (Fig.4F and Fig.6F), which confirmed that complex I deficient muscle had increased complex IV.

8. Figure S3 - the authors conclude there is increased mitochondrial proliferation but it is important to evaluate these samples for markers of mitochondrial fragmentation as the EM images suggest this may also be occurring. EM data would be stronger if quantified.

We agree that without quantification we should refrain from making statements on mitochondrial numbers from the EM images. Therefore, we removed those statements from the text (Page 7). Later in the manuscript, quantitative data showed an increase in mitochondrial markers. Mitochondrial fragmentation was also addressed in 6 months old gastrocnemius mouse samples. See below the answer and figure in “General point 2” answer.

9. Figure 3 - Please indicate what dose was used in vg/kg for the different treatment times.

Figure 3B has been updated to include this information: 1.25×10^{15} vg/Kg. We have also included the vg/kg in the methods section of the manuscript as requested.

10. Figure 4 - The authors describe the treatment as showing restored NDUFS3 protein levels in all muscle homogenates analyzed. Please show these data for other muscles and quantify as done earlier in the manuscript. Why does one male appear to express far more GFP than the others and why does there appear to be more NDUFS3 expression in treated females than males?

To satisfy the Reviewer’s request we have now included the analyses of gastrocnemius (Fig. S5 and S6) and quantified the western-blot. In addition, we analyzed several other muscle groups, including: TA, diaphragm, eye and arm muscles in some animals (Fig. S5, S6), and these ones were only analyzed in few animals therefore the quantification is not shown. We have also analyzed nonskeletal muscle tissues, including: heart, liver, and brain and quantified protein expression (Fig. S5 and S6). The quantification of NDUFS3 expression in quadriceps is shown in figure 4 (young injected mice) and 6 (adult injected mice). Regarding the question of why did we observe different levels of GFP or NDUFS3 in different mice (males and females), we believe it is related with the efficiency of the injection and/or to the intrinsic and individual variability normally observed among experimental mice.

11. Figure 4 - Why are only females included in 4B and C? Please include both males and females and quantify the data. D-F are these males or females? - please separate and quantify.

The males were included in a supplementary figure (Fig. S4) as the results were identical to the ones observed with the females. As requested, we have now included the male data in the main Figure 4 for SDS-PAGE, Blue Native PAGE and Blue Native- In-gel activity assays. The quantification of females is still in Fig. 4, whereas the quantification for males is shown in Supplementary Fig. S4C-D.

12. Figure S4 and S5 - please switch these as they are called out 5 then 4 in the text.

Thank you for detecting this mistake. The text has been corrected.

13. Figure S5 - this does not represent AAV9 distribution. Please present vg data for tissues to show AAV9 distribution.

We agree with the reviewer. We have now changed the figure to provide a better description of the data. We have also expressed the vg copies for both treatment groups, the young-injected mice (Fig. S5) and adult-injected mice groups (Fig. S6). We have also added the methods information in the appropriate section.

14. Figure S5 - Why does panel A show a combination of 6 and 15 month data and B is only 6 month data? Please provide replicates, quantify all of these data, and present in a similar format across the injection times for all information in S5.

We clarified the figure. Now, Panel A represents mice injected at 15-18 days and sacrificed at 6 months. Mice sacrificed at 15 months are now shown in panels E-F. The control is still mice sacrificed at 6 months because these mice do not leave more than 6-8 months, in contrast to the mice injected with AAV9-Ndufs3. Quantifications are provided in Fig S5B. The analysis of more mice for additional tissues such as the brain, heart, liver and gastrocnemius have also been included in figure S5.

15. Figure 5 - Please separate these data between males and females and provide an explanation for what has changed between 5 and 5.5 months for 5C. Again - sample sizes are low for these types of data.

The mice behaviors were evaluated before injection at 2 months and showed significant signs of myopathy regardless of the gender (Fig. S4A-B). Therefore, these data were not separated. The reasons are: 1) myopathy onset was similar for both sexes, 2) we have a relatively small number of animals due to high titer AAV9 required to inject, 3) The marked/significant difference between treated and untreated mice, regardless of sex.

What changed between 5 and 5.5 months? We are not sure we understood the question well, but we believe the Reviewer was referring to the fact that at 5.5 months the hang test reached significance whereas at 5 months it did not, even though there is an almost identical pattern. The 5.5 mice were actually tested right before sacrificing at 6 months, so they were closer to 6 months. To better reflect that, we changed the description in the figure to 6 months. Again, mouse behavior tests are variable, and the small differences between the tests at 5 and 6 months are not surprising. The hang test showed the same results for both 5 and 6 months, even if significance was reached only for the 6 months group.

Per the reviewer's request we did try to increase the number of injected animals for this symptomatic group of mice as a larger sample size would be better. That is the reason why it took us 6 months to resubmit. To do so, we re-ordered the recombinant AAV viral prep from the same core we obtained the original prep (University of Iowa Viral Core Facility). However, the AAV9-Ndufs3 viral titer obtained this time was 2.6×10^{12} vg/mL. We remind that the titers of the original prep were 3.1×10^{14} vg/mL (two logs higher). The core guarantees (for \$3,000/large high titer prep) a titer of $> 10^{12}$ vg/mL, so we could not complain. Nonetheless, we ordered another batch, hoping for a titer similar to the original one. The next batch titer was 8.1×10^{12} vg/mL. Again, within the core guarantee, but way below the 10^{14} titer obtained and used in the initial experiments. Even if still low, we decided to inject a small group of 2-month old symptomatic mice with AAV9-Ndufs3 (n=4). We injected as much volume as we could retro-orbitally (150 μ L). Not surprisingly, we could not detect expression of NDUF3 in the muscles of treated animals injected with these titers 4 months after injections (negative data available upon request). Although we understand that time and financial constraints are not reasons to not do an experiment, we believe that the initial data we have, even with a small "n" is of high magnitude, uncontroversial and significant enough to leave no doubt that the result is real. Although the requirement for high rAAV titers for in vivo expression is well known, we have emphasized this point by including a paragraph in the results section describing the lack of expression when lower titers were used.

16. Figure 5D - Please separate males and females.

As mentioned above, we believe it is not necessary to separate males and females for this experiment because the onset of myopathy was shown to be similar for both sexes and the phenotypic recovery is equally marked and significant.

17. Figure 5F - please quantify these data across multiple sections from multiple mice - the NDUFS3 treated group looks to have fewer positive fibers than WT... please explain. Is there evidence of fiber type switching in this model with or without treatment?

The COX/SDH activity staining was performed simultaneously in all mice groups, and COX and SDH were stronger in KO-GFP samples, compared to WT or KONDUFS3 mice samples. We chose not to quantify these data for the following reasons: 1) The method is intrinsically semi-quantitative, 2) Small differences in section thickness promote changes in staining intensity, 3) Even though it shows obvious differences, the levels of mitochondrial proteins have been quantified and better demonstrated that the smKO mice display increase mitochondrial levels in muscle. These levels are restored to “WT-like” upon AAV9-mediated NDUFS3 replacement. We did measure and quantified complex IV in-gel activities for the KO-GFP mice samples which was significantly increased when compared to the KO-NDUFS3 and WT samples. We have replaced the histology panel (Fig.5F) with images at lower (10X) magnification to provide a larger field that better exemplifies the color intensity of the fibers.

As suggested by the reviewer, we addressed fiber switching by RT-qPCR analysis using different primers for slow and fast fiber types (MHCI, MHCIIa, MHCIIb, MHCIIx). No significant changes were observed, as exemplified below (Fig. R1) for group A.

Figure R1. Change of MHC I/MHC II in the young mouse group.

18. Figure 6A, B, C - [please quantify results.

#The quantification of the panels was included.

19. Methods - Vector description needs to include an actual dose/kg for injections.

This information has been included in the methods and figures.

20. Discussion - Please discuss cause of death in this model.

We have not studied the cause of the death in this model; therefore, we can only speculate. We added the following statement to the beginning of the Discussion: "...” The cause of death was not determined, but mice were profoundly emaciated and unable to move normally around the cage”

21. As the authors list only 2 human patients with some form of NDUFS3 disorder and mention other groups who have shown gene therapy type improvements in other mitochondrial genetic disorders, please provide more discussion on how this study is innovative.

We have included a new patient reference, from 2018. In total, there are only 3 patients with mutations in the NDUFS3 gene, all of them are missense point mutations.

In terms of innovation, we are presenting a novel and unique mouse model (isolated complex I defect in muscle) and a therapy approach that has not been described before (reversion of a myopathy by restoring complex I). To our knowledge there are very few publications on gene therapy of mitochondrial diseases, which include mostly studies from the Zeviani/Viscomi group.

The closest is:

AAV9-based gene therapy partially ameliorates the clinical phenotype of a mouse model of Leigh syndrome. Di Meo I, Marchet S, Lamperti C, Zeviani M, Viscomi C. In this study a CNS predominant model was used to treat with rAAV9 NDUFS4. Different model and different target tissue. The conclusions were: “We found that IV administration alone was only able to correct the cI deficiency in peripheral organs, whereas ICV administration partially corrected the deficiency in the brain. However, both treatments failed to improve the clinical phenotype or to prolong the lifespan of *Ndufs4*^{-/-} mice. In contrast, combined IV and ICV treatments resulted, along with increased cI activity, in the amelioration of the rotarod performance and in a significant prolongation of the lifespan.”

The differences between this and our study are obvious.

There are 4 additional publications from the Zeviani/Viscomi group showing that rAAV9 expression of MPV17, OPA1, ETHE1 and TYMP that improve models of mitochondrial disease, but none of which is similar to our study as they either used a mouse model that is not deficient in a single OXPHOS defect, or used a therapy that was not a replacement of the missing gene. Finally, there is a study from the Wallace group (ref 48) showing a replacement of ANT1, which is not an OXPHOS complex, focally (intra-muscle injection).

We believe that, among the different aspects of this study, showing that skeletal muscle can remain viable for 2 months not having complex I, and that upon gene replacement function is restored to close to normal, are innovative observations.

22. Discussion page 2 - the authors say the 2 month injections are post-symptomatic but that may not be true for females based on the data as presented - when male/female data are separated out it will be potentially possible to draw these conclusions.

Both males and females were symptomatic at 2 months. Please see figure S4 for a separation of females and males' behavior data at 2 months. Both males and females were symptomatic at 2 months.

23. Discussion page 3 - The authors state that AAV is non-pathogenic but this is clearly a hotly debated topic in the field. Please address these issues and other limitations.

Although there could be some consequences of using AAV for gene therapy, it is universally accepted to be the best tool available at the moment. To satisfy the reviewer, we removed the word “non-pathogenic” from the Discussion (it was mentioned only once, page 17).

Other General Comments:

1. It is important to include vg and RNA transcript data in addition to protein data in studies such as this so that mechanistic conclusions regarding the level of regulation for these interacting proteins can be drawn.

The vg analysis is now included in Fig. S5 and S6, as requested. We believe that the protein data is crucial for the message of our manuscript, therefore, we have not evaluated RNA transcript data.

2. Mitochondrial proliferation is mentioned several times but often mitochondrial fragmentation is a huge issue for muscle related mitochondrial disorders so evaluating these markers will provide a more complete picture of the impact of this deficiency.

As requested by the Reviewer, we have analyzed mitochondrial fusion (OPA1, mitofusin2) and fission (DRP1) markers by western-blot. We have found that there were no significant differences in

mitochondrial fragmentation (fission) markers.

Drp1 was relatively decreased in the smKO compared to OpaI and mitofusin 2, reflecting the active mitochondrial proliferation. We did find that the small form of OPA1 tended to be increased in the smKO mice. However, we do not see how this could fit in our manuscript at this point, as it is a new study on its own, which requires further exploration. We also found an increase in mitofusin 2. We are adding these results below (Fig.R2) and if requested, we can incorporate it in the manuscript, but again, we believe it would be just a distraction from the main message.

Figure R2. Analysis of mitochondrial dynamics in 6-month-old gastrocnemius samples. (A) Comparison of WT^{flx}, KO-NDUFS3 and KO-GFP protein expression of OPA1, MFN2 (fusion markers) and total DRP1 (fission marker) in gastrocnemius samples of P15-18 injected mice. n=3-4 mice per group. MFN2- mitofusin2, ns- not significant. (B) Western blot quantification of (A). (C) Same as (A) but in P60 injected mice. n=3-4 per group. (D) Western-blot quantification of (C). Data was normalized with the stain-free loading. Statistical analysis was performed with One-way ANOVA followed by Bonferroni post-test, p<0.05 was considered significant.

3. All the characterizations performed in the initial figures should be repeated in the treated groups - EM, 2 month treated body weights for male and female curves over time and centralized nuclei measurements.

We did not perform EM analyses or central nuclei determinations (except for the young injected mice, quantification in Fig. S4F), but do not believe they are required to validate our conclusions. In addition, the body weight curves for male and female are shown for the young mice injected group (Fig. 3D and E). For the adult mice injected group, as previously mentioned, it was not possible to separate females from males, therefore we are not showing the body weight curves.

4. This is a mitochondrial story so providing oxygen consumption data would greatly enhance your study by providing direct mitochondrial functional data...all mitochondrial researchers reading this in its current state would be left disappointed and wondering what oxygen consumption would look like in these animals.

Although mitochondrial respiration measurements could have been performed, we did obtain functional OXPHOS data in the form of complex I (in vitro and in-gel activity) and complex IV in-gel activity. Because the main goal was to recover complex I levels and activity, we believe that the results were clear and oxygen consumption experiments, which would require a new set of AAV9-treated animals (fresh tissue) would not be necessary.

2nd Editorial Decision

19 November 2019

Thank you for the submission of your revised manuscript to EMBO Molecular Medicine. We have now received the enclosed report from the referee who was asked to re-assess it. As you will see the reviewer is now fully supportive and I am very pleased to inform you that we will be able to accept your manuscript pending minor editorial amendments.

I look forward to reading a new revised version of your manuscript as soon as possible and within 2 weeks.

***** Reviewer's comments *****

Referee #3 (Comments on Novelty/Model System for Author):

The manuscript is greatly improved.

Referee #3 (Remarks for Author):

Your effort towards addressing my comments is noted and your work has resulted in a stronger manuscript.

2nd Revision - authors' response

4 December 2019

Authors made the requested editorial changes.

Corresponding Author Name: Carlos T. Moraes

Journal Submitted to: EMBO MOL MED

Manuscript Number: EMM-2019-10674-V2